# Systems that support hearing families with deaf children: A scoping review

**Julia Terry** [1]*, **Jaynie Rance**[2]

**1** School of Health and Social Care, Faculty of Medicine Health and Life Science, Swansea University, Wales, United Kingdom, **2** School of Psychology, Faculty of Medicine Health and Life Science, Swansea University, Wales, United Kingdom

* j.terry@swansea.ac.uk

## Abstract

### Background

Over 90% of deaf children are born to hearing parents who have limited knowledge about deafness and require comprehensive support and information to support and communicate with their deaf child. However, little is known about the systems that support hearing families with deaf children. We performed a scoping review to provide an overview of current literature on the topic.

### Methods

The protocol of the scoping review was prepared using the PRISMA statement guidelines for scoping reviews. Relevant search terms were used to identify eligible studies following discussion with the study's steering group. Databases searched were CINAHL, Medline, ProQuest Central and ASSIA, as well as grey literature from relevant journals and online sources. Included were studies published from 2000 to 2021 and available in English.

### Results

A search of databases identified 1274 articles. After excluding duplicates, screening titles and abstracts and full texts, 65 papers matched the identified inclusion criteria. Results included 1 RCT, 7 comparative studies, 6 literature reviews, 4 PhD theses, and 47 further empirical studies.

### Conclusion

There is limited quality evidence on what supports hearing parents with deaf children. It is evident that further studies are needed to ensure comprehensive support is accessible and effective for hearing parents of deaf children.

**Data Availability Statement:** All relevant data are within the paper and its Supporting information files.

**Funding:** This scoping review and the linked descriptive qualitative study are part of the

SUPERSTAR project funded by Research Capacity Building Collaboration (RCBC) Wales, a Welsh Government funded scheme through Health and Care Research Wales, which exists to increase research capacity in nursing, midwifery, the allied health professions and pharmacists across Wales. The SUPERSTAR project is a Postdoctoral Fellowship, where the funder provides access to a supervisor, and a Community of Scholars to support and promote high research quality and outputs. The funder had no role in study design, data collection and analysis, decision to publish, or preparation of the manuscript.

**Competing interests:** The authors have declared that no competing interests exist.

# Introduction

## Authors' note

In this paper the terms Deaf and deaf are used. A capital D for Deaf is used to refer to people who identify as Deaf and view themselves as part of Deaf communities, are a Deaf adult, Deaf professional or Deaf mentor, or who may be profoundly Deaf and may use a signed language. When a lower-case d for deaf is used this tends to refer to deaf children or those who are hard of hearing. Currently there is limited consensus about an emic term, as people can feel colonised when a specific label is provided and may be in different places in their individual journey [1].

Over 5% of the world's population experience deafness or hearing loss [2] and by 2050 hearing loss will affect one in ten people. Currently there are an estimated 34 million deaf children globally [3], and nearly 55,000 deaf children in the UK [4]. As 96% of deaf babies are born to hearing parents [5, 6] who are usually not expecting to raise a deaf child, it is important that families benefit from a range of support processes and interventions. Support in this context can best be described as encouragement, help and enablement, to promote sustainable success and confidence for hearing parents and their deaf children.

When parents find out their child has been diagnosed as deaf or having hearing loss, or when they suspect this to be the case, families begin a journey that involves differing amounts of support, information, and guidance. For many families, initial discussions begin at newborn hearing screening, if these services are available. Newborn hearing screening has become an essential part of neonatal care in high-income countries with positive outcomes following early intervention during the critical period to enable optimal language development. Currently at least 45 US states require new-born hearing screening by law [7] and others have achieved this without legislation or have it pending. In the UK the NHS newborn hearing screening programme recommends screening for all babies in the first five weeks of life, although there is a notable absence of hearing screening in the Global South [8, 9]. The early detection of hearing status can prevent significant detrimental effects on cognitive development happening later. For example, if children's development needs are not fully addressed [10] a deaf child may not develop language skills to ensure fluent communication as a vital platform for further learning. Language deprivation in the first five years of life appears to have permanent consequences for long-term neurological development [11].

Whilst families welcome prompt hearing screening, it is worth bearing in mind the range of perspectives that exist about deafness. Parents say they encounter predominantly medical model approaches, which suggest their child has a deficit [12], proposing that deafness is treated and seen as an impairment [13]. Hearing families may find later that there are cultural-linguistic models and alternative approaches that help them understand the social identity of their deaf children. The socio-cultural view that considers the rich environment of Deaf communities, including the naturalness of sign languages with deafness seen as a way of being, and not an impairment [14]. Diagnostic rituals can set in motion a deficit-orientated way of addressing a child's needs, sometimes resulting in diminishing parental competence and confidence [15]. Often parents report that initial information received upon early detection of their child's hearing loss can be incomplete and coloured by workers' personal beliefs and values, usually originating from a medical model [16], when healthcare policies could acknowledge the broad scope of conflicting views that hearing parents may encounter.

Hearing screening, identification and individualised early intervention is critical in helping deaf or hard of hearing children achieve their full potential [17] and has led many nations to develop Early Hearing Detection and Intervention (EHDI) programs. It may be audiology, speech and language services or education professionals who begin to provide parents with advice about communication choices and pathways. Frequently the not-for-profit or charity

sector agencies provide additional support and information perhaps because they have wider scope in terms of delivery arrangements.

Systems that support hearing parents with deaf children may include education, health, care, and social services, depending on the child's age and location. Support may be provided by statutory services and the voluntary sector and may include short-term initiatives and long-term input. Essentially the support families have and the advice they are given in the early years of their child's life is of key importance. Hearing parents will want to know about how the ear works, about deafness, communication and language choices, their child's emotional and social development, education, alerting and assistive devices as well as early years support. At an early point there will be discussions with the family about the child's language development and communication options. Professionals who support families with deaf children may hold a range of views towards sign language, but essentially families will decide about communication choices and whether their child will learn a mixture of spoken and signed language or just a spoken language [18]. Decisions made about communication choice will likely affect the child and family for a lifetime [19].

Fully accessible language experiences during the early years are vital in empowering deaf children's development potential [20]. There is a critical window for language development and if a child is not fluent in a language by around the age of five years old [21], he or she may not achieve full fluency in any language. It is a foundational language that is key to the development of future language. Sign language often comes naturally to deaf children, and deaf children exposed to sign language during the first 6 months of life have age-expected vocabulary growth when compared to hearing children [22–24], meaning that learning a signed language can avoid language delays. If parents are keen for their deaf child to learn speech, then sign language does not impede this. Parents can be given misinformation and not be made aware that there are risks in excluding sign language during the critical time of language acquisition, with no evidence that sign language causes harm [25]. There are recommendations for changes in existing systems to support bimodal bilingualism as default practice, in order to provide the best educational outcomes, which means a signed language and a spoken language [23]. It is suggested that all deaf children should be bilingual [26]. However, little is known about the support parents are given at the outset of these decision-making processes.

Critics suggest there is a need to stop dichotomizing spoken or signed language, and to focus instead on educating families about the range of opportunities available [19, 27]. Frequently hearing parents of deaf children do not know where to turn for support and can be overwhelmed with advice as they try to understand different methods employed in the language development and education of their child [20]. Support for hearing parents of deaf children varies globally. A variety of initiatives and projects appear regularly in local and regional news stories, such as support for sign language classes [28], family camps for deaf children [29] and artificial intelligence avatars that help deaf children to read [30]. Support systems are people or structures in society that provide information, resources, encouragement, practical assistance, and emotional strength.

We argue that there is limited published evidence about the support systems for hearing parents with deaf children. Therefore, we conducted a scoping review to provide a baseline overview of the published evidence until 2021 of the extent, variety, and nature of literature in this area.

## Aims of the study

The aim of this scoping review was to map available evidence regarding the systems and structures surrounding deaf children and their families with hearing parents/guardians.

The specific objectives were to:

1. identify published studies describing support systems and structures that support hearing parents with deaf children, and

2. review the evidence of these studies.

The primary objective of this review was to assess the number of studies and their characteristics such as their origin, study designs, study population, type of support and key findings regarding systems or supports for hearing parents with deaf children.

## Methods

### Study design

We followed PRISMA-ScR guidelines for scoping reviews in the conduct of the literature review, data extraction/charting, and synthesis. The main aim of a scoping review is to identify and map the available evidence for a specific topic area [31]. The approach to the review was based on Arksey and O'Malley's framework [32] which consists of the following stages: i) identifying the research question; ii) identifying relevant studies; iii) selecting studies; iv) charting the data; and v) collating, summarising and reporting the results. Ethical approval was not required because the study retrieved and synthesised data from already published studies.

### Identifying the research question

The core aim of this scoping review was: What is the existing research that examines support systems for hearing parents with deaf children. The focus on hearing parents was due to over 90% of deaf children being born to hearing parents, who have little knowledge of deafness and deaf people, which is different from the experience of Deaf parents parenting deaf children [33, 34]. An initial a priori protocol was developed and published on *Open Science Framework* in February 2021, and then revised using feedback from the project steering group over the course of the project, as scoping reviews are an iterative process [35]. The steering group comprised Deaf and hearing professionals and lay members, people working with Deaf charities, in health, education, policy and academia. Decisions were documented in a search log and steering group meeting notes to record the scoping review process. The final protocol was registered on 24th August 2022 with the Open Science Framework—https://osf.io/w48gc/.

### Identifying relevant studies

The scoping review research question was left intentionally broad and was discussed in-depth at the first project steering group where members generated 50 words and terms to be included in the outline database searches. The evidence was searched using four electronic databases, hand searches of reference lists of key journals and repositories (such as PROSPERO), and contact made with key authors; as well as internet site searches for policies and reports. The wider project involves interviewing family members and workers situated in Wales, UK, so the scoping review included material specific to Wales as well as other geographical areas nationally and internationally that has contextual similarities (for example, grey literature including newspaper articles about family situations and support projects, blogs and regional reports), and these were included in the early stages of the review. An experienced information specialist's help was sought in reviewing the PICO framework (see Table 1) and specific search strategies. The databases included were CINAHL, Medline, ASSIA and Proquest Central, with searches conducted between May and June 2021, and updated in January 2022. (An example of the search strategy for one database is provided as an additional file).

**Table 1. PICO framework.**

| PICO elements | Keywords | Search terms | Search strategies |
|---|---|---|---|
| P (Patient or Population) | Hearing parents, Guardians, Family members, care givers, Primary carers, Families, | Parents/families | famil* or relative* or parent* or sibling* OR guardian* OR 'care giver*' OR carer* OR 'caregiver*' |
| | Deaf/deaf/ hard of hearing/ DHH/ impairment | Deaf | AND |
| | Sign/Signed language/BSL | children | deaf* or hard of hearing or hearing impaired or d/hh OR Deaf* |
| | | | OR d/Deaf OR D/deaf |
| | Child/children/youth/adolescents/ teenagers | | |
| | | | AND |
| | | | Sign* OR sign* language OR BSL |
| | | | AND |
| | | | Child or adolescen* or youth or children or teen* OR young people |
| I (Intervention or Issue) | Development, communication, | Social, emotional, language development, communication | social OR emotional OR development OR language OR education OR communication OR intervention OR speech OR pathway OR referral OR diagnosis OR decision making OR process |
| | Intervention | Intervention, pathway, referral | |
| C (Comparison of intervention) | N/A | N/A | N/A |
| O (Outcome | What helps support hearing parents, what are the challenges/barriers/ facilitators | Support, systems | Support OR system OR community OR help OR challenges OR barriers OR facilitators OR information OR choice OR assist OR service |

Different techniques and terms were used to expand and narrow searches, including tools such as medical subject headings (MESH), Boolean operators and Truncation. Single and combined search terms included key subject area on deafness, children, BSL/sign language and parent/family words. Limitations were set to include papers in the English Language and peer-reviewed research from the time period January 2000 onwards. In addition, key journals, professional organisation websites and reference lists of key studies were searched to identify further relevant documents. The final search strategy and terms were agreed and verified by a health subject librarian.

Inclusion criteria were: published research articles and dissertations, literature reviews and PhD theses specific to a) parents and families/caregivers b) deafness/hard of hearing/hearing loss c) sign language or British Sign Language (BSL) d) child or young person e) information specific to support, systems, challenges, barriers f) were published in English between 2000–2021. The inclusion criteria were purposely broad, as there is a dearth of scientific evidence on the area of support and systems for hearing parents with deaf children.

Exclusion criteria were: papers pre-2000 (unless they met a-e of inclusion criteria above); papers without a focus on deafness, papers that focused solely on literacy, or were short news items or opinion papers, and/or did not focus on support issues for hearing parents of deaf children.

## Study selection

The initial search produced a total of 1274 results from database searches (see PRISMA, Fig 1), which were screened, and a further 192 records were added from internet and hand-searching. An example of a database search is provided in Table 2. Once duplicates were removed (n = 2653 +18) and a further 8 discounted as pre-2000 that did not meet the inclusion criteria, 1202 publications remained, and titles and abstracts were screened. 821 records were then removed in line

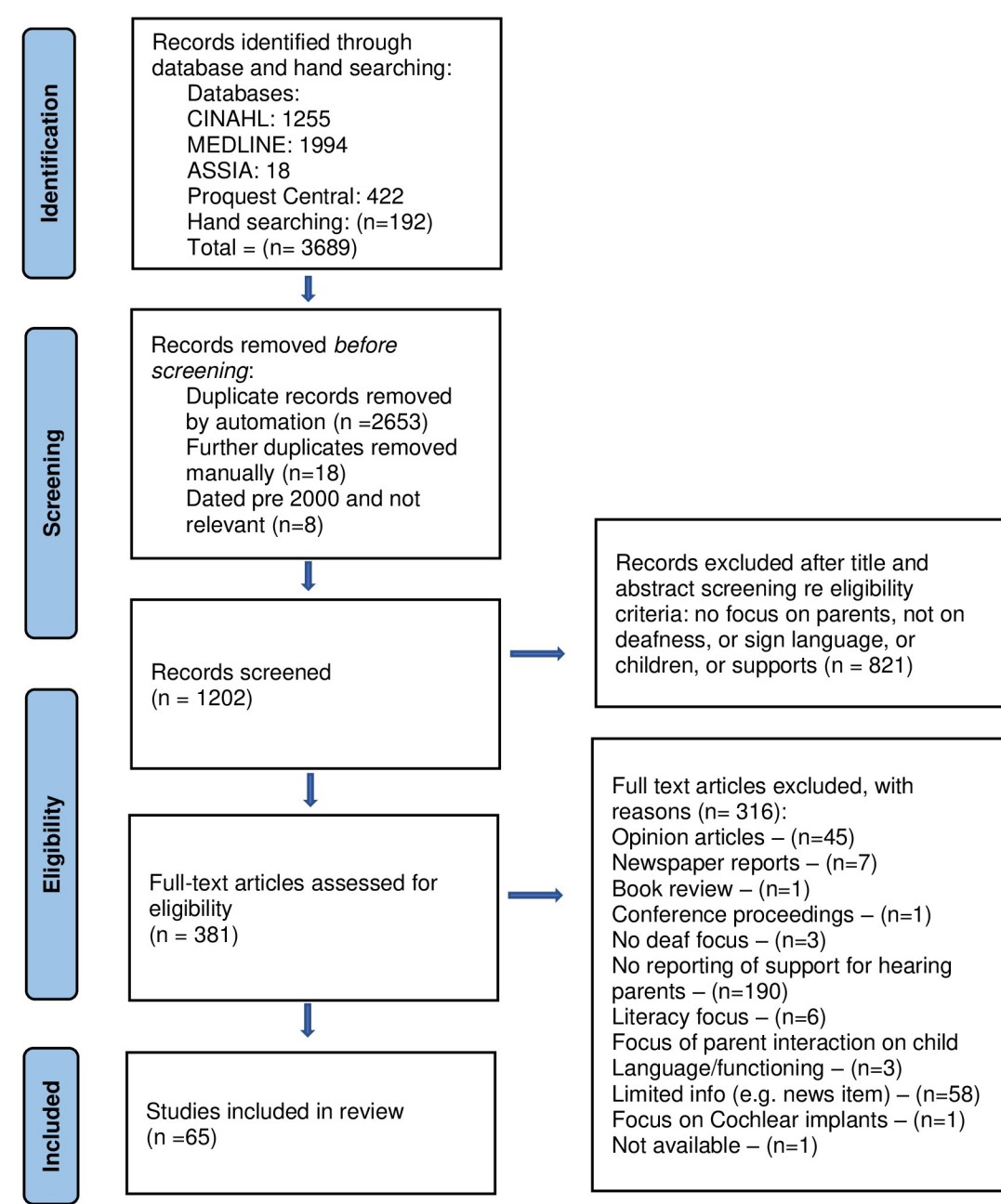

**Fig 1. PRISMA diagram for the scoping review process.**

with the eligibility criteria, and the remaining 381 full texts were obtained, and details transferred to an Excel database for sifting. Knowledge synthesis was achieved by peer review using Rayyan software [36] and annotated spreadsheets of retrieved papers, which were reviewed by two researchers independently with inter-rater discrepancies resolved by discussion.

We began by excluding sources that did not describe support for hearing parents of deaf children, such as opinion articles, newspaper reports, and papers without a deaf focus. Screening full texts resulted in a further 316 papers being excluded, leaving a total of 65 publications included in this review (see Fig 1 PRISMA diagram).

**Table 2. Example of one database search.**

| Cinahl | | |
|---|---|---|
| via EBSCOhost | | |
| Search date: 07/09/2022 | | |
| Records identified: 1255 | | |
| #1 | (famil* or relative* or parent* or sibling*) OR guardian* OR 'care giver*' OR carer* OR caregiver*. | (874,306) |
| #2 | (MH "Family+") | (264,527) |
| #3 | (MH "Extended Family+") | (5,581) |
| #4 | (MH "Parents of Children with Disabilities") OR (MH "Single Parent") | (7,069) |
| #5 | (MH "Siblings") | (6,556) |
| #6 | (MH "Foster Parents") | (1,081) |
| #7 | '''care giver''' | (25,755) |
| #8 | S1 OR S2 OR S3 OR S4 OR S5 OR S6 OR S7 | (937,221) |
| #9 | (deaf* or hard of hearing or hearing impaired or d/hh) OR d/Deaf OR D/deaf OR Deaf* | (16,123) |
| #10 | (MH "Deaf-Blind Disorders+") OR (MH "Deaf Education") | (1,841) |
| #11 | (MH "Hearing Loss, Partial+") OR (MH "Hearing Screening") | (13,415) |
| #12 | (MH "Deafness+") | (8,541) |
| #13 | S9 OR S10 OR S11 OR S12 | (27,709) |
| #14 | sign* OR sign* language OR British Sign Language OR BSL | (1,451,905) |
| #15 | (MH "Sign Language") | (2,273) |
| #16 | (child or adolescen* or youth or children or teen*) OR young people | (1,178,696) |
| #17 | (MH "Child+") | (748,420) |
| #18 | (MH "Adolescence+") | (587,821) |
| #19 | (MH "Children with Disabilities") | (13,309) |
| #20 | S16 OR S17 OR S18 OR S19 | (1,301,016) |
| #21 | S14 OR S15 | (1,451,905) |
| #22 | S8 AND S13 AND S20 AND S21 | (1,255) |

## Charting the data

A data-charting form was developed by one reviewer, and then updated iteratively in discussion with the second reviewer, which was piloted and found to be effective. The data extracted were the author, year of publication and country of origin, study design, sample population, study aim and findings and study strengths and weaknesses, (see Table 3). Articles meeting inclusion criteria were examined, and data was entered into Excel spreadsheets, which included sample characteristics (age range, clinical characteristics, sample size); and experimental and control measures, as applicable.

Through this process sources were identified as follows: 55 primary research studies, four PhD theses and six literature reviews.

## Collating, summarizing and reporting results

From the final scoping review, 21 individual countries were represented (Fig 2, which present the distribution by country). Most publications came out of the USA, Australia, the UK and Canada, which may be due to greater funding in this area of research compared to other nations.

Due to the heterogeneity of the range of study contexts, a narrative synthesis was a reasonable way to approach the reporting of retrieved studies. After summarising the information

**Table 3. Characteristics of included studies for SUPERSTAR scoping review.**

| Study, year, location | Theme | Study design | Sample population | Study aim | Findings | Strengths | Weaknesses |
|---|---|---|---|---|---|---|---|
| 1. Ahmad & Brown, 2016, Australia | Communication choices and strategies | Comparative: Questionnaires on individual communication strategies and 3 min mother-child interactions videoed and analysed how strategies utilised | 16 mothers—allocated 8 mothers of D/HH children diagnosed in last 18 months and 8 experienced mothers with D/HH children diagnosed more than 24mths | To explore whether duration and type of early intervention (EI) involvement affect the value parents place on intervention strategies (difference between new and experienced hearing mothers with deaf children) | Only minor differences related to time spent in Early intervention programs, so EI programs do not necessarily bring about changes to parents' knowledge or resultant communication strategies | Provides insight into what parents think are important strategies and what parents actually do. Ethical approval specified; informed consent achieved | Only a small pool of available participants. Variables chosen may be insufficient to capture change. Parent/child may have been interacting for some time before EI program |
| 2. Alfano, 2019, Latin America | Communication choices and strategies | Other—qualitative, ethnographic interviews and participant observation | Data analysed from 12 mother/grandmother interviews, 12 child interviews and 12 observations, recruited through agencies in Southeast USA | To identify how Hispanic mothers communicate with their children with hearing loss who use ASL as their primary language | Few mothers learned ASL or did ASL in Spanish so language early on was limited, as many mothers had not learned until child was older, and mostly used oral communication | Identified many issues that need targeting to improve communication | All mothers were Hispanic, over 66% of children were male, 66% mothers had more than high school education. Findings may not translate for less educated parents, and older children |
| 3. Baker & Scott, 2016, USA | Interventions and resources | Other—qualitative Longitudinal case study including records, and participant and teacher interviews | Case study on one Latina student | To provide a longitudinal case study of one deaf Latina student about their educational experiences from high school to graduation | Recommendations for placement, early communication needs and techniques are highlighted. Teachers would like to see more education for families to help them understand services they are entitled to. Intensive language immersion is necessary to develop a strong L1 base/speaker's first language | Illustrates need for instructional strategies for Deaf multi-lingual learners | Focus is on one individual. Assessments used were designed for monolingual hearing children. Interviewing teachers who knew participant at early years stage was not possible |
| 4. Beatrijs et al., 2019, Belgium | Communication choices and strategies | Comparative: cross-sectional longitudinal study. Parents recruited from home-based early intervention team | First, 1 Deaf and 2 hearing mothers, interactions with deaf children recorded over 18mth. Second, interactions of 5 mothers and 5 fathers with their deaf children were analysed for strategy use | To identify which strategies deaf and hearing parents prefer and implement in their daily communication with their deaf children | Deaf parents outperformed the hearing parents in the duration of successful interaction moments with their deaf children. Deaf parents are best positioned to inform hearing parents on visual communication, language and Deaf culture | Results display trend and importance of visual communication. Added value that fathers included as participants | Small sample as difficulty reported reaching parents, children's exposure to much early testing and low number of parent willing to participate |

*(Continued)*

**Table 3.** (Continued)

| Study, year, location | Theme | Study design | Sample population | Study aim | Findings | Strengths | Weaknesses |
|---|---|---|---|---|---|---|---|
| 5. Behl et al., 2017, USA | Interventions and resources | Comparative<br><br>Deaf and hard of hearing children assigned to early intervention via either telepractice or face to face for a six-month intervention period which focused on coaching caregivers to enhance language development | 48 deaf and hard of hearing children and their families, and 15 providers from 5 early intervention programs | To compare the outcomes of telepractice to traditional in-person services to families with deaf and hard of hearing children (DHH).<br><br>Specifically (1) How do families and their children who are DHH who receive services via telepractice compared to those who receive services via in-person visits in regard to child and family outcomes? | Supports the effectiveness of telepractice in delivering early intervention services to families of children who are deaf or hard of hearing.<br><br>Telepractice is a useful tool in early intervention in delivering family-centred services | Verifies that telepractice can support the development of deaf and hard of hearing infants and their families | Randomisation to telepractice or in-person group was partial, as real-world constraints were considered e.g. travel. A larger sample would have increased statistical power. Child development measures were administered by early intervention provider, not an objective tester. A more culturally and linguistically diverse population could have been sought |
| 6. Blaiser et al., 2013, USA | Interventions and resources | Randomised controlled trial<br><br>Pre and post-test measures of child outcomes, family and provider satisfaction and costs for six-month intervention period | 27 families with infants with varying levels of hearing loss | To compare the costs and effects of TI compared to traditional in-person early intervention service delivery. The study engaged providers and families from a state-wide early intervention program for infants and toddlers who were DHH | The tele-intervention group scored significantly higher on expressive language measure and parent engagement<br><br>Tele-intervention is a promising cost-effective method for delivering high quality early intervention services to families of children who are DHH | Use of a comprehensive measure of the quality of intervention | Sample size small and duration of study were short. Degree of training and staff skillset need further exploration |
| 7. Bortfeld & Oghalai, 2020, USA | Communication choices and strategies | Comparative | Hearing parents with their hearing children (n = 4) and hearing parents with their deaf children (n = 4) | To characterize establishment of joint attention in hearing parent—deaf child dyads and hearing parent—hearing child dyads | Joint attention as an indicator of early communicative efficacy in parent—child interaction for different child populations. There is an active role parents and children play in communication, regardless of their hearing status. Joint attention helps language development indirectly | Interactive behaviours, regardless of hearing status, might be tracked over time. Knowing association between joint attention and successful language development, understanding parent accommodation of deaf children's unique communication needs is important | Small sample. More date required to understand relevant factors |

(*Continued*)

**Table 3.** (Continued)

| Study, year, location | Theme | Study design | Sample population | Study aim | Findings | Strengths | Weaknesses |
|---|---|---|---|---|---|---|---|
| 8. Borum, 2012, USA | Communication choices and strategies | Other—qualitative, in-depth interviews | 14 African American parents of deaf children, recruited from two schools for the Deaf in Washington, D.C., USA | To explore parents' perceptions of communication choice | Professionals need to understand cultural ecology in relation to communication choices. Families relied more on native spoken language (English) and fingerspelling | Findings similar to Gerner de Garcia (1993) [114] and Steinberg and Davila (1997) [115] | Not generalisable to all African American families with deaf children |
| 9. Bruin & Nevøy, 2014, Norway | Communication choices and strategies | Other—qualitative, initial demographic survey then analysis of written personal accounts | 27 written parent personal accounts | To examine discourse on communication modality on experiences with follow-up after paediatric CI constructed | Parents' choice of communication modality is demanding, characterized by insecurity and will continue to be so. Therefore, families need follow-up systems that can support them in negotiating the various options available | Study provides insight into understanding of discourse modalities and to address need for increased awareness on how discourse governs parents' and professionals' thinking | Need to for further research with different backgrounds, including those who provide support |
| 10. Carey-Sargeant & Brown, 2003, Australia | Interventions and resources | Comparative<br><br>Videotaped free play sessions with language samples transcribed, coded and analysed | 12 mother-toddler dyads (toddler 25–45 months): 6 hearing mothers and their 6 deaf toddlers and 6 hearing mothers and their 6 hearing toddlers | To examine 'pausing' during interactions between profoundly deaf toddlers and their hearing mothers compared to hearing mother-toddler dyads | Pausing plays a significant relationship in the development of interactions and use between deaf child-hearing mother dyads is different to use between hearing child-hearing mother dyads. Greater proportion of longer pauses in deaf child-hearing mother group<br><br>Speech pathologists, audiologists and teachers of the deaf may benefit from adding pause features to their interactions and to other early intervention programs | Confirms previous research findings about benefits of pausing during communication interactions | Small sample; all from middle-class English-speaking families |

*(Continued)*

**Table 3.** (Continued)

| Study, year, location | Theme | Study design | Sample population | Study aim | Findings | Strengths | Weaknesses |
|---|---|---|---|---|---|---|---|
| 11. Crowe & McLeod, 2014, Australia | Communication choices and strategies | Other—quantitative, translational summaries of 4 studies investigating communication choices of children (n = 406), parent/teacher questionnaires, audiology information, questionnaires on parent decision making who were participating in the Longitudinal Outcomes of Children with Hearing Impairment (LOCHI) study in Australia. | Children with hearing loss (n = 406) and their parents (n = 792) | To examine the communication usage of young Australian children with hearing lost and to explore factors that influence parents' decisions of communication mode | Parents reported professional advice about access to audition, interventions, and opportunities | Research summary paper targeted at professionals working with children with hearing loss | Data relates solely to demographics of communication style choices for child, mother and father, and other languages known —summary paper |
| 12. Crowe et al, 2012, Australia (same study as 13) | Communication choices and strategies | Other—quantitative, data for 406 children collected (communication mode, oral language, demographics) | Records of 406 hard of hearing children collected through LOCHI study (population-based data of audiological, speech, language outcomes in New South Wales, Queensland and Victoria) | To investigate the communication and language use of a population sample of 3-year olds with hearing loss and their caregivers | No relationship was found between the caregivers' hearing status and the children's communication mode. Significant association between presence of disability in addition to hearing loss and communication mode used by children aged 3 | Provides initial examination of cultural and linguistic diversity and heritage language attrition of population | 19.9% study population lived in disadvantaged decile in Australia, so harder to recruit and maintain in study. Difficulties in distinguishing type of sign language uses. Absence of information about children's language proficiency, so diversity of language not captured |
| 13. Crowe et al., 2014, Australia (two papers on same study 12 and 13) | Communication choices and strategies | Other—qualitative, questionnaire | 177 caregivers (of 157 Australian children with hearing loss) | To explore the decision-making factors regarding use of speech, sign and multilingualism | The advice of speech-language pathologists, audiologists, and specialist teachers were more important to caregivers than advice from medical practitioners and non-professionals | Broad exploratory view of influences on caregiver decision making | Questionnaire provided finite list of influences, so some may have been omitted. Design did not allow for dynamic nature of caregiver decision making |

*(Continued)*

**Table 3.** (Continued)

| Study, year, location | Theme | Study design | Sample population | Study aim | Findings | Strengths | Weaknesses |
|---|---|---|---|---|---|---|---|
| 14. Dammeyer et al., 2018, Denmark | Family perspectives and environments | Other—quantitative survey method | Sixty-five Danish children with cochlear implants (CIs) aged 11–15 years were asked about their CI use and other factors related to communication, experiences of hearing loss, social participation and friendships, and psychological well-being. | To explore the perspectives and concerns of 11-15-year-olds with Cochlear implants (Cis) | Findings raise cause for concern regarding some children with CIs. The subjective experience of hearing loss and CIs, loneliness, and feelings of difference suggest that some young people with CIs struggle with self-concept issues<br><br>Education and support for children with CIs should be planned and tailored to diverse needs. Such planning needs to include their perspectives, not just those of their parents and teachers | Sample focused on young people | Small sample. Children with severe social, psychological, and cognitive difficulties may be less likely to participate. Findings may not be generalizable to countries that do not have free healthcare for all |
| 15. Davids et al., 2018, Multi | Interventions and resources | Other—Literature review of nine databases, retrieving five studies reporting on intervention programs with a focus on parenting styles | N/A | To identify and evaluate previous research on intervention programs that focus on providing support for parenting styles | Summaries of each of the five studies are presented with reach, efficacy, adoption, implementation and maintenance. | Review highlighted important challenges and strengths that clarify significance of intervention for hearing parents with deaf children. Parent benefits and child outcomes are included | Greater participation by females than males in intervention programs, so more specialised intervention programs targeting fathers could be developed |
| 16. Decker et al., 2012, USA | Communication choices and strategies | Other—quantitative, online survey on knowledge of communication development, choices of communication method and what they felt influenced their decision | Parents with children with hearing loss under 7 years, n = 34 | To explore additional influences on parents' choices of communication for their deaf child | Results indicated no effects of parents' knowledge of development on their communication choices, but did indicate an effect of parents' values and priorities for their children No group differences in sources parents cited as influential; all parents relied on their own judgment | Recruitment from non-biased organisation which incorporated diversity | Participants all drawn from one source and incorporated children in broad range of ages, all parents were Caucasian with high levels of education, so not generalizable. Small sample |

*(Continued)*

**Table 3.** (Continued)

| Study, year, location | Theme | Study design | Sample population | Study aim | Findings | Strengths | Weaknesses |
|---|---|---|---|---|---|---|---|
| 17. Edelist, 2019, Canada<br><br>PhD Thesis | Communication choices and strategies | Other—qualitative: analysis of infant hearing program documents and parent interviews | 12 parents of deaf children (aged one to ten years, Ontario-based | To examine how language and deafness are made meaningful through text and lived experience, and how parents come to make hearing technology and communication modality choices for their children amongst competing discourses of deafness and language | Discourse of screening implies hearing levels as problematic, with deafness as an unthinkable outcome, and spoken language as 'right' way forward.<br><br>Parents may resist medical knowledges of deafness and request alternate services as they get to know their child beyond diagnostic assumptions. Findings indicate parents and their children may be better aided by services that promote a wider variety of communication options. Comprehensive information about sign language and Deaf culture not included in hearing program | Exploration of ways infant hearing programs can imagine deafness as something other than a problem | Sample limited to hearing parents, difficult to recruit deaf parents to study |
| 18. Eleweke & Rodda, 2000, UK | Communication choices and strategies | Other—qualitative, case studies | Case studies of two families, recruited from Audiology, Manchester, England | To examine factors contributing to parents' selection of a communication mode to use with their children with hearing loss | The factors influencing parental choice were grouped under four themes: (a) the influence of information provided to parents, (b) parents' perceptions of assistive technology, (c) attitudes of service professionals and educational authorities, and (d) quality and availability of support services | Underlines the need for core support services for parents of deaf children | Small sample of two families |

(*Continued*)

**Table 3.** (Continued)

| Study, year, location | Theme | Study design | Sample population | Study aim | Findings | Strengths | Weaknesses |
|---|---|---|---|---|---|---|---|
| 19. Flaherty, 2015, Australia | Family perspectives and environments | Other—qualitative<br><br>Semi-structured interviews | 18 hearing parents of deaf children in Western Australia (WA), recruited through WA Deaf Society | To examine the experiences of hearing parents of deaf children spanning various life stages. | Themes included: trauma of diagnosis of deafness, model of Deafness, Australian Sign Language, Cochlear implant, needs of the child at various life stages. Themes offers insight for professionals. How hearing parents are helped to understand deafness and the support they receive may influence not only their child, but their own and their family's health. Deaf parents of deaf children have much to offer hearing parents of deaf children | Reflexive rigour used by researcher. Life-grid used to aid memory recall | Potential for recall bias as events discussed relied on parents' retrospective memories |
| 20. Friedman Narr & Kemmery, 2015, USA | Interventions and resources | Other—qualitative<br><br>Examining parent mentors' summaries of conversations with more than 1,000 individual families of deaf and hard-of-hearing (DHH) children receiving parent-to-parent support as part of an existing family support project | Database of 1056 families of deaf and hard-of-hearing (DHH) children receiving parent-to-parent support<br><br>Data from 5150 excerpts of conversations with mentors | To explore parent mentors' summaries of conversations with more than 1,000 individual families of deaf and hard-of-hearing (DHH) children receiving parent-to-parent support as part of an existing family support project. | Three topics were the most prevalent within the conversations between parent mentors and family members: hearing-related topics, early intervention, and multiple disabilities. Several differences emerged between English-speaking and Spanish-speaking families receiving support. | Large sample, and parent mentors' perspectives not analysed before. Potential for this study to impact policy issues pertaining to the crucial need for parent-to-parent support within deaf education and early intervention programs | Findings are based upon subjective notes written by five parent mentors after speaking with families. A large volume of data was coded *by excerpt* rather than being coded *by family*. If the data were initially coded by family, some of the findings could have been correlated with other findings |

(*Continued*)

**Table 3.** (Continued)

| Study, year, location | Theme | Study design | Sample population | Study aim | Findings | Strengths | Weaknesses |
|---|---|---|---|---|---|---|---|
| 21. Gale et al., 2021, Multi (USA, Europe, South America, Asia, Africa and Australia | Interventions and resources | Other—quantitative<br><br>Online survey | 48 respondents completed the survey | To investigate roles of Deaf adults in early intervention programs with deaf children. | Support provided by deaf adults in Family Centred Early Intervention programs includes educational information and communication support, and that major roles provided by deaf adults are as role models and language providers. Additionally, respondents reported families do not have a diverse range of deaf professionals to connect with in early intervention programs. There is a need to infuse deaf adults in programs that include Formalisation, Collaboration, Education, and Infusion. | Validated earlier anecdotal evidence about role of Deaf adults in early intervention programs globally | 85 participants (from 133) started the survey but did not complete as may have had no intervention or first point of contact to mention. Survey was only in English and online. |

(*Continued*)

**Table 3.** (Continued)

| Study, year, location | Theme | Study design | Sample population | Study aim | Findings | Strengths | Weaknesses |
|---|---|---|---|---|---|---|---|
| 22. Hadjikakou & Nikolaraizi, 2008, Cyprus | Family perspectives and environments | Other– Semi-structured interviews on personal communication memories | 24 deaf individuals | To investigate personal communication memories of Deaf adults when they were children in their families. | Those who graduated from the school for the deaf, and used sign language from an early age had negative communication experiences at home. It was found that they could not achieve communication either in CSL or orally with their hearing parents (n = 12). On the other hand, those participants who graduated from general schools did not record any negative memories, since they could communicate from an early age with their hearing parents through speech (n = 10). Similarly, the two participants, who attended the school for the deaf, and signed with their family Deaf 1 members from an early age, described pleasant communication memories<br><br>An early and mutual mode of communication between families and deaf children ensures good communication and experiences. | Highlights the importance of an early and mutual mode of communication between family members and their deaf children, regardless of the communication modality | Participants were asked to re-count stories of their childhood regarding their communication experiences which may entail risks (e.g. re-structuring memory). Only one participant with Deaf parents participated in the study. Given the small numbers of deaf children with Deaf parents (5– 10%) and that only 24 participants were enrolled, the small number of Deaf participants with Deaf family members can be understood |

**Table 3.** (*Continued*)

| Study, year, location | Theme | Study design | Sample population | Study aim | Findings | Strengths | Weaknesses |
|---|---|---|---|---|---|---|---|
| 23. Hadjikakou & Nikolaraizi, 2011, Multi | Family perspectives and environments | Other—qualitative<br><br>Semi-structured interviews | 24 Cypriot (12 men, 12 women) and 22 Greek (12 men, 10 women) Deaf individuals | To explore the current functions of Deaf clubs in Greece and Cyprus | Deaf clubs in both countries provide a gathering place for deaf people, organize social and sport activities, and promote their demands through legislation. In addition. Deaf clubs maintain and transmit Deaf culture and history to future generations, offer Deaf role models to young deaf children | Deaf clubs have strong presence in two countries studied so findings may be applicable to other countries with Deaf clubs. | Limited sample from two small countries |
| 24. Hall et al., 2018, USA | Family perspectives and environments | Other—quantitative<br><br>Data analysed from the Rochester Deaf Health Survey±2013 (n = 211 deaf adults) for associations between sociodemographic factors including parental hearing status, and recalled access to childhood indirect family communication | 211 Deaf individuals' existing data from the Rochester Deaf Health Survey-2013 (RDHS-2013).<br><br>The University of Rochester IRB determined the RDHS-2013 to be surveillance and not research records | To assess the influence of parental hearing status on deaf people's recalled access to childhood indirect family communication | Deaf people who have hearing parents were more likely to report limited access to contextual learning opportunities during childhood.<br><br>Parental hearing status and early childhood language experiences, therefore, require further investigation as possible social determinants of health to develop interventions that improve lifelong health and social outcomes of the underserved deaf population | Findings exemplify the 'dinner table syndrome' phenomenon that is a widespread experience for deaf people, but has yet to be studied analytically | Recruitment methods mainly focused on outreach to deaf sign language users. No measure of childhood communication modalities (e.g. spoken language, sign language) was included and childhood experience was based on recall |
| 25. Hardin et al., 2014, USA | Communication choices and strategies | Other—qualitative, focus groups | 9 parents and 1 professional participants (all female, 6 Deaf, 4 hearing)<br><br>ASL users recruited from South-eastern US urban area | Aim of this focus group study was to understand the experiences of families who chose ASL as their communication mode. | Findings show a need for continued professional development about the complexities of self-identity, Deaf culture and modes of communication for families with ASL users | All participants shared integral role of ASL in their identity and culture | Lack of diversity in participants, small sample |

(*Continued*)

**Table 3.** (Continued)

| Study, year, location | Theme | Study design | Sample population | Study aim | Findings | Strengths | Weaknesses |
|---|---|---|---|---|---|---|---|
| 26. Henderson et al, 2016, Multi | Interventions and resources | Other - Delphi study, mixed methods<br><br>eDelphi methodology with quantitative and qualitative elements | 31 experts selected from nine countries, with panel diversity | To guide the development of a conceptual framework<br><br>Dual stage project: i) scoping review then ii) stakeholder consultation via questionnaires round 1 and round 2 | Increased understanding of the role of parent-to-parent support in Early Hearing Detection and Intervention (EHDI) programs<br><br>The conceptual framework demonstrates the centrality of parent-to-parent support in EHDI | International representation, heterogeneity of participants. Integration of peer-reviewed literature and expert representation addressed academic, tacit and experiential knowledge for this framework | Due to closed questions and English as a second language for some participants, it was difficult to ensure a quantitative consensus. However, using qualitative data preferences could usually be determined. Regional preferences for labels play a part |
| 27. Hintermair, 2006, Germany | Family perspectives and environments | Other—quantitative<br><br>Questionnaires included PSI, SDQ, SOC, F-SozU | 213 mothers and 213 fathers of deaf and hard of hearing children | To examine the correlation between parental resources, sociodemographic variables, parental stress experience, and child socioemotional problems | High parental stress is associated with frequent socioemotional problems in the children, emphasizing the importance of a resource-oriented consulting and support strategy in early intervention because parental access to personal and social resources is associated with significantly lower stress experience. Child development seems to profit enormously from a resource-oriented support concept.<br><br>Additionally, results confirm earlier findings: parents with children with additional needs are especially stressed and the child's communicative competence are a better prediction than linguistic medium (spoken language or sign) | Large sample. Aware of importance of need of individualised approach with families | Need further longitudinal research including younger children. Some concepts in this study (cf. specific parental competence, specific social support, child's communicative competence) may show deficits because of the relatively few items used to assess them |

(*Continued*)

**Table 3.** (Continued)

| Study, year, location | Theme | Study design | Sample population | Study aim | Findings | Strengths | Weaknesses |
|---|---|---|---|---|---|---|---|
| 28. Frush Holt et al., 2012, USA | Family perspectives and environments | Other—quantitative<br><br>Self-report family environment questionnaire (Family Environment Scale | Forty-five families of children with cochlear implants | To examine the social climate of the family, relationships and growth in families with a young CI user. | Family environments can be modified by therapy and education to maximise support and children's development<br><br>Families who perceive themselves as organized reported their children experienced fewer problems related to inhibitory control. Families with high levels of organization place importance on structure and planning in family activities and individual responsibilities within the home. These beliefs provide a mechanism for maintenance of the coherence of the family system (similar to findings from Coldwell, Pike, & Dunn, 2006 [116]; Hughes & Ensor, 2009) [117] | Focus was on a neglected domain of study that deserves further attention | Study relied solely on parent reports of executive function and family environment. Need multi-method, multi-trait longitudinal research design using parent reports combined with performance measures of executive function |

*(Continued)*

**Table 3.** (Continued)

| Study, year, location | Theme | Study design | Sample population | Study aim | Findings | Strengths | Weaknesses |
|---|---|---|---|---|---|---|---|
| 29. Huang, 2017, USA | Deaf dichotomies | Other—qualitative<br><br>In-depth interviews | Case study excerpts from two sets of siblings n the Deaf bilingual-bicultural (Bi-Bi) community. | To explore case studies of potential consequences of language loss, and related social and cultural experiences. | Misconceptions about bi- and multilingualism often lead healthcare professionals to recommend that parents limit deaf children to learning oral English only. Preventing the child's exposure to the home language and culture could result in severe, long-term consequences in the child's development. On the contrary, a child's knowledge of multiple languages and cultures can result in fluid conversational exchanges, trusting parent-child relationships, and strong cultural identity. clinical implications for clinicians supporting Deaf families in the healthcare system include recruiting ASL medical interpreters; providing written resources in plain, simple language; researching the Deaf culture's social behaviours and communication style; and learning how cultural differences affect communication about healthcare needs | Highlights communication as a human right and importance of cultural competence for clinicians | Limited basis for generalization of results to the wider population |

(*Continued*)

**Table 3.** (Continued)

| Study, year, location | Theme | Study design | Sample population | Study aim | Findings | Strengths | Weaknesses |
|---|---|---|---|---|---|---|---|
| 30. Huiracocha-Tutiven et al., 2017, Equador | Family perspectives and environments | Other—qualitative<br><br>Interviews | 9 principal carers of deaf children (parents/grandparents) of DHI children in Ecuador | To explore how socioeconomic and cultural factors influence the experiences of hearing parents of deaf children. | Many parents are critical of the way schooling has been available for their children, and are worried about discrimination | Little is known about the experiences of DHI children and their parents in the fields of education and employment in Ecuador. Development of policies aimed at promoting good quality education, career opportunities, and financial independence of people with disability requires further investigation of the current situation | Audiograms (with or without an assistive device) is an inadequate indicator of an individual's ability to function effectively in one or other context. Moreover, the assessment procedure and the entitlement to benefits that it determines is seen as unfair and (at the margins) arbitrary However, this study suggests a number of steps that clearly need to be taken |
| 31. Johnson, 2015, USA<br><br>PhD thesis | Deaf dichotomies | Other—qualitative<br><br>Participant observation and interviews over one academic year | Seven preschoolers (four and five-year olds) with deafness or speech delay, six hearing mothers, four teaching staff | To explore relationship between identity processes and multimodal interactions in daily focal events among four and five-year-old deaf and hard of hearing children and their parents, teachers and hearing peers in a California preschool | The "hearing mom-deaf child" experience offers a window into understanding language and cultural practices in unconventional ways and complexifies notions of being socialized in and through language. Students, teachers and mothers drew on semiotic resources to position themselves and their interlocutors indexing insiderness or outsiderness to the larger ideological constructs of hearingness and deafness undermining structural notions of the Deaf identity | Offers insights for broadening the scope of the how we begin to conceptualize the role of modality in interaction and the experiences of the deaf learner, understudied in the larger Second Language Acquisition narrative | Explored the use of modality in one American preschool classroom with a small group of Students, and was confined to analysing structured focal events |

(*Continued*)

**Table 3.** (Continued)

| Study, year, location | Theme | Study design | Sample population | Study aim | Findings | Strengths | Weaknesses |
|---|---|---|---|---|---|---|---|
| 32. Kisch, 2008, Holland, Israel | Deaf dichotomies | Other—qualitative<br><br>Observation, anthropologicial | Al-Sayyid Arab-Bedouin shared signing community in Negev, Israel | To explore a Bedouin shared-signing community and advocates closer investigation of both facilitating and disabling social practices, which would also allow better examination of comparable cases | Deaf people became capable members of their community, well equipped with social networks and resources to cope with social challenges within the community and beyond<br><br>Deaf signing communities have been observed to have distinct rules for attention getting, turn taking, polite discourse, joking, name giving, and other behaviours related to language | Attempted to delineate a communication web comprised of numerous language modes and domains that involve constant translation and development of alternative channels of communication and indirect access to information | Focus on one specific culture. Further constitutive factors and social dynamics that constitute deafness in comparable cases need to be considered |
| 33. Levesque et al., 2014, Australia | Communication choices and strategies | Other—qualitative<br><br>Single case study | Single case study on one Australian boy at age 23 and then 42 months | To investigate the communication and language development of a deaf boy over a 20-month period, specifically the bimodal bilingual input. Data collected bi-monthly<br><br>Part of larger study of 8 deaf children | English and Auslan vocabulary growth over time was strongly correlated with the parents' sensitivity to his communication needs | Valuable insight into the developmental path taken by a deaf child as he determines language modality best suited to his needs | Small sample of a single case |
| 34. Luckner & Muir, 2001, USA | Communication choices and strategies | Other—qualitative, semi-structured interviews and classroom observations | 20 deaf students, 13 teachers of the deaf, 19 general education teachers, 19 parents, 9 interpreters, 2 notetakers | To examine factors that contributed to Deaf students' success in a general education setting | Identified ten themes for success which included family involvement, early identification and collaboration among service providers | Themes can be used to guide family discussion in determining needs | Acknowledge that factors will vary according to individual and setting |
| 35. Luckner & Velaski, 2004, USA | Family perspectives and environments | Other—qualitative<br><br>Interviews (email—11, phone—7 & face to face—1) | 19 families with deaf children participated<br><br>Recruited through Teachers of deaf children in one Western state asked to nominate one healthy family with a deaf child | To identify factors that families believe contribute to family health where children are deaf. | Insights shared by families will better position other families and professionals to form partnerships to help children who are deaf. | Insights shared by families will better position families and professionals to form partnerships to help children who are deaf and hard of hearing to communicate, think independently and master knowledge and skills for surviving and thriving | Small sample as all 19 families lived in one state, 18 out of 19 were white |

(*Continued*)

**Table 3.** (Continued)

| Study, year, location | Theme | Study design | Sample population | Study aim | Findings | Strengths | Weaknesses |
|---|---|---|---|---|---|---|---|
| 36. Maluleke et al., 2021, South Africa<br><br>Lit review | Interventions and resources | Other—Literature review of four databases between 2009–2019 reporting on family centred early intervention programs for children who are DHH<br><br>22 peer-review research studies included | N/A | To explore and document current evidence reflecting trends in FCEI for children who are deaf or hard of hearing (DHH) by identifying and describing current practice models and/ or processes of FCEI for these children | Findings were discussed under 5 themes: caregiver involvement; caregiver coaching/ information sharing; caregiver satisfaction; challenges with FCEI; and telehealth. Generally, there is sufficient evidence for FCEI, with caregivers indicating the need for full involvement in their children's care | Findings provide a springboard for the implementation and evaluation of FCEI programs, especially in the South African context | Limitations of the review are not stated |
| 37. Mapp & Hudson, 1997, USA | Family perspectives and environments | Other—quantitative<br><br>Questionnaire on resources and stress (Friedrich, Greenberg & Crnic, 1983 [118]) | 98 parents of children with hearing loss who attend a private school for deaf and hard of hearing<br><br>53% were boys, 47% were girls All aged 3 to 14 years | To determine the stress and coping responses of African American and Hispanic families that included a deaf or hard of hearing child. | The ability through signing was significantly related to the stress.<br><br>The ability of the child to communicate by signing was found to be significantly related to the level of stress experienced by the parents or caretakers. Parents whose child signed well or fluently indicated significantly less stress<br><br>One unexpected finding was the low level of stress expressed by the sample | Hispanics are more likely to use the strategies of distancing, self-control, seeking social support, planful problem solving, and positive reappraisal than African Americans. Hispanics are more likely to use confrontive and planful problem solving strategies than Whites and Asian Americans. | Parents in this study had already adjusted to knowledge of their child's disability may have mitigated the initial stress reaction. |
| 38. Marschark et al., 2012, Multi | Family perspectives and environments | Comparative<br><br>Questionnaire—children's and parents' perceptions of academic functioning linked to social functioning | 54 deaf children and 54 hearing children from both USA and UK, aged 5–12 years | To explore perspectives on academic and social aspects of children's school experiences were obtained from deaf and hearing children and their (deaf or hearing) parents | Deaf children having deaf parents, attending a school for the deaf and using sign language at home all were associated with more positive perceptions of social success. Use of cochlear implants was not associated with perceptions of greater academic or social success. | Perceptions of social success among deaf children, in contrast, are significantly affected by whether they have deaf or hearing parents, whether parent—child communication involves sign language and children's school placements | It is not known whether perceptions are borne out by actual evaluations of social interactions and networks |

(*Continued*)

**Table 3.** (*Continued*)

| Study, year, location | Theme | Study design | Sample population | Study aim | Findings | Strengths | Weaknesses |
|---|---|---|---|---|---|---|---|
| 39. Matthijs et al., 2017, Belgium | Deaf dichotomies | Other—qualitative<br><br>In-depth interviews about raising a deaf child | Three hearing Flemish mothers with deaf children who had been exposed to discourses about medical model and cultural-linguistic model of Deafness | To investigate three Flemish mothers' engagement with educational options for their child | Findings showed alternative explanations for former findings concerning mothers' decision-making processes, especially the difficulty of learning sign language as a second language in an effort to provide a bilingual—bicultural education and highlighted the importance of having rich experiences. | Positioning theory offered a particular lens that yielded valuable insights | Small sample. Lens from positioning theory used, recognise other researchers' conclusions may have differed |
| 40. Napier et al., 2007, Australia | Interventions and resources | Other—qualitative<br><br>Focus group to develop new curriculum<br><br>Survey of professionals<br><br>Parent consultation | Numbers not included in paper | To discuss the design of a new curriculum to teach sign language to hearing parents with deaf children<br><br>Focus of paper is to share process of curriculum development within an action research framework | Implementation of the curriculum confirmed a lack of resources, leading to further research and the development of family-specific resources for teaching and learning Auslan. | Overview paper of process of engaging networks and development of Auslan for families' resource. The process of development of these resources has potential application for other signed language teachers, researchers, teachers of the deaf and associated professionals who are working with families in their learning of a signed language | Limited detail about specific elements of project |
| 41. Page et al., 2018, USA | Interventions and resources | Other—quantitative<br><br>Questionnaire, interviews—part of longitudinal study<br><br>Children's hearing, speech, and language data were collected from annual testing and analysed in relation to service data | Participants included parents of CHH (preschool n = 174; school n = 155) and professionals (preschool n = 133; school n = 104 | To examine the service setting, amount, and configuration and analysed the relationship between service receipt and student hearing levels and language scores | A majority (81%) of preschool-age CHH received services. Children were more likely to be in a preschool for children who are deaf or hard of hearing (CDHH) or exceptional children than a general education preschool. By elementary school, 70% received services, nearly all in general education settings. | Findings support the need for increased implementation of interprofessional practice among SLPs and teachers of CDHH, as well as audiologists, to best meet the needs unique to this population | Participants with limited knowledge or training would be less likely to report confidence especially if their experience of working with CDHH children is limited |

(*Continued*)

**Table 3.** (Continued)

| Study, year, location | Theme | Study design | Sample population | Study aim | Findings | Strengths | Weaknesses |
|---|---|---|---|---|---|---|---|
| 42. Park & Yoon, 2018, Korea | Family perspectives and environments | Other—qualitative | Five Korean mothers of oral deaf children aged 16–19 years | To understand the nature and aspects of the parenting stress that Korean mothers of deaf children experience to shed light on the development of social services and policies that empower parents to enhance their deaf child's psychosocial development in South Korea. | Researchers constructed 21 concepts and 7 categories, the categories being "Frustration with parenting their child," "Struggling between mainstream education and special education," "Continuing to be alienated from mainstream education settings," "Feeling left out and hurt in family relationships," "Making a sacrifice for the child," "Change in values of life," and "Importance of services meeting parents' needs." The study suggests the need for comprehensive support services that consider deaf children and their parents, siblings, families, and schools. The study also provides clinical implications for social work practice with families with deaf children | It is likely that there are some common experiences that mothers of deaf children undergo that span the cultural differences between Korea and America. This study offers meaningful contributions to social work practice with families with deaf children from culturally diverse backgrounds | Factors beyond child's deafness may have affected parents' stress level: socioeconomic status, educational background, the child's academic achievement, and early intervention. These variables were not included as selection criteria—nor were they collected—because the main focus of the study was the general and common experiences of hearing mothers with deaf children who attended a mainstream school in South Korea |

**Table 3.** (Continued)

| Study, year, location | Theme | Study design | Sample population | Study aim | Findings | Strengths | Weaknesses |
|---|---|---|---|---|---|---|---|
| 43. Pfister, 2017, Mexico | Family perspectives and environments | Other—qualitative<br><br>Ethnographic field project—participant observation | 39 parents/ guardians of deaf children in one school in Central Mexico City (participants were mostly hearing) | To illustrate that Mexico's therapeutic approach to language does not constitute language socialization for deaf children; simultaneously, it affirms that signing communities offer sites where deaf people can actively engage in this critical process | Artificial language environments of the therapeutic approach do not produce language acquisition for all deaf participants, and, even those who were relatively successful in oralist environments could not participate fully in socialization processes in a predominantly hearing, mainstream society | Children (or novices) are exposed to language and culture through their social participation in a community; second, that children are socialized naturally as they are repeatedly exposed to norms, expectations, and social connectivity through language narratives of these participants illustrate how the contrived environment of language therapy did not facilitate either of these two processes<br>The variation represented by ethnographic examples contrasts with standardized expectations that all deaf children can achieve similar outcomes based upon oralist goals and medicalized intervention. | Sample from one demographic area |

**Table 3.** (Continued)

| Study, year, location | Theme | Study design | Sample population | Study aim | Findings | Strengths | Weaknesses |
|---|---|---|---|---|---|---|---|
| 44. Pfister, 2018, Mexico | Family perspectives and environments | Other—qualitative<br><br>Interviews | Over thirty interviews with parents of deaf children | To explore the features of Mexican families' complex journeys ('pilgrimages') as they coped with the "predicament" of childhood deafness | Parents individually reached the conclusion that their children were not inherently "disabled." As they learned that a quick "fix" or "cure" for their children's hearing was a false promise, the primary goal became finding communication. In other words, as they realized how the narrow perceptions of deafness in medicalized settings often created the most significant barriers to parents' primary desire for their children, they moved from a medical fix to a linguistic fix. | Ethnographic data presented here emphasize the transformative experience of these journeys despite prevailing medicalized paradigms and tragedy tropes. Keeping in mind that "deaf identities are learned through practice in social contexts, depending on the cultural resources available" (De Clerck 2009:151) [119], these findings highlight Mexican families' agency and perseverance as they escaped the hegemonic structure of medical authority and chose different cultural characteristics on behalf of their deaf children. | May not be transferable to other areas |
| 45. Pizer et al., 2007, USA | Communication choices and strategies | Other—qualitative, case studies | Three central Texas famiies | To examine how three baby-signing families use signs in their daily lives. | Baby signing fits into the parenting ideologies present in the professional class in USA. | Analysis of social context of baby signing and role of signing in interactions | Small sample, naturalistic research with cameras on parents and children would allow more analysis |
| 46. Richardson, 2014, USA | Communication choices and strategies | Other—Literature review: using terms 'deaf and healthcare' and 'deaf and culture'<br><br>Leininger's theory of cultural care diversity used as theoretical framework to evaluate articles | N/A | Three factors affect health behaviours: health literacy, culture/cultural barriers, and language proficiency. This paper discusses them in relation to Deaf culture | Imperative that healthcare providers ensure they provide culturally competent care and their practices accommodate for deaf patient needs to ensure equitable care and positive health outcomes | Ethnic components of Deaf culture identified. Recommendations included to improve patient/provider communication | Literature not explicitly presented in terms of amount or studies |

(*Continued*)

**Table 3.** (Continued)

| Study, year, location | Theme | Study design | Sample population | Study aim | Findings | Strengths | Weaknesses |
|---|---|---|---|---|---|---|---|
| 47. Sajjad et al., 2016, Pakistan | Family perspectives and environments | Other—qualitative<br><br>Interviewed with structured questionnaire | Fifty hearing parents of deaf children aged up to 10 years approached through five special schools in Karachi City | To explore reactions from parents about having a deaf child, and to explore strategies, behaviour and needs. | Most parents said they wanted counselling in domains including; assessment and diagnosis of hearing impairment, communication strategies with hearing impaired children, speech therapy, hearing aid maintenance and dealing problems of hearing impaired children.<br><br>Creating awareness about the importance of counselling sessions and designing structured counselling programs for parents of hearing-impaired children at suitable venues like hospitals, schools or from the platform of any association | Three-pronged approach is needed including; educating parents about the child's amplification system, parents counselling, and parents' training, in line with other studies' findings | Limited details of results as % only shown |
| 48. Scarinci et al., 2018, Australia | Communication choices and strategies | Other—qualitative, semi-structured interviews | 7 caregivers with children with hearing loss in Australia (6 in Victoria, 1 in New South Wales) | To explore caregiver decisions about communication method | The family unit is at the core of decision making and has important clinical implications regarding early intervention professionals' provision of family-centred services when working with the families of children with hearing loss | In-depth interviews covered entire communication journey | Participants' memories may be influenced by outcomes and successes resulting in recollection bias. Multilingualism and Deaf culture were not explored. Need wider variety of cultural backgrounds |
| 49. Sisia, 2012, USA<br><br>Phd thesis | Family perspectives and environments | Other—Qualitative<br><br>Phenomenological study of two deaf adolescents from hearing families | Two deaf adolescents from hearing families who attend school in the mainstream academic setting, and their parents | A phenomenological research study that shares the experiences of two adolescents from hearing families who attend school in the mainstream academic setting | Additional research is needed to assist mainstream educators and districts in their incorporation of students with a hearing loss into their classrooms | The level of hearing loss and the primary mode of communication employed has a significant bearing on which culture participants chose to identify with. Current absence in the literature about modern technology, social networking and its influence on the social experience of the deaf | Small sample |

*(Continued)*

**Table 3.** (Continued)

| Study, year, location | Theme | Study design | Sample population | Study aim | Findings | Strengths | Weaknesses |
|---|---|---|---|---|---|---|---|
| 50. Snoddon & Underwood, 2014, Canada | Deaf dichotomies | Other—qualitative<br><br>Individual and focus group interviews with parents and instructors<br><br>Shared reading program– 13 workshops over ten months reading books in ASL | Two families, both with three-year-old deaf children; two Deaf instructors | To address recognized gaps in early intervention programming in terms of bilingual bicultural ASL and English services | There remain overt and obvious barriers to inclusion such as lack of access to ASL in schools and communities, and an absence of Deaf community oversight of schools and agencies serving Deaf people<br><br>As young Deaf children become 'readers of power' and come to understand how adults, including parents, are themselves 'implicated' in the early intervention context that casts ASL-using people as other, the possibility emerges for children and parents alike to directly challenge the structures and ideology at play | Supporting ASL community programs as integral sites of early intervention and education for Deaf children. Such approaches work to nurture the social relationships and capabilities that shape the lived experiences and futures of individual Deaf children | Small sample |
| 51. Steinberg et al., 2003, USA | Communication choices and strategies | Other—qualitative, questionnaire and semi-structured interview | 29 Hispanic families recruited from four areas: Pennsylvania, Texas, central Florida and northern California, recruited by community facilitators | To explore the impact of language, culture, minority status, and access to information and services on the decision-making process | The communication method chosen tended to be the one recommended by professionals, usually a combination of spoken English and sign language. Parents frequently expressed the hope that their child would learn Spanish as well. These subjects displayed a higher degree of assertiveness in obtaining services for their children than other studies have suggested. | Findings underscore need for greater presence of Hispanic professionals, educators and peer-to-peer liaison supports to ensure higher degree of cultural competence | Small self-selected sample. Parents with older deaf children may not recall events accurately from when their child was young compared to parents with younger children |

(*Continued*)

**Table 3.** (Continued)

| Study, year, location | Theme | Study design | Sample population | Study aim | Findings | Strengths | Weaknesses |
|---|---|---|---|---|---|---|---|
| 52. Storbeck & Calvert-Evers, 2008, South Africa | Interventions and resources | Other—Overview of implementation of home-based early intervention project in South Africa | 37 families registered on program | Exploring the pilot implementation of a home-based early intervention support. Aim of HI HOPES is to partner with parents, informing and equipping them in their journey with their infant with a hearing loss, | There is a need for effective organization and communication between primary health care practices (both public and private) and intervention agencies to improve coordination and implementation of services. | Seeks to improve the way early intervention services are organized and delivered for all families with d/hh infants with particular attention to prevention outcomes for low income families of d/hh infants | Lack of detailed data |
| 53. Takala et al., 2000, Finland | Interventions and resources | Other—quantitative  A 49-item questionnaire—annual  Study of 'A Good Future for Deaf Children'–a 5-year educational project  Once weekly, each child met with a teacher who was deaf. Parents, siblings, and other relatives met about once monthly to study sign language, and all families in the project signed together about twice yearly | 81 families with deaf children in Finland, with 52 boys and 35 girls | To understand how children learned to sign  Study addressed four questions asked of parents about the project: (a) How did the children learn to sign? (b) Did both the parents and the children benefit from the project? (c) What was the position of sign language in the family? (d) Did the project have some impact on the family's social network? | Families indicated satisfaction with the project; they learned to sign and their social networks expanded. Parents favoured bilingual education: Sign language was the main language but learning Finnish was also important  Learning sign language was not easy, especially for the fathers. The families that were most actively involved in the lessons learned the most. | Longitudinal study over time with high responses as questionnaire administered at annual event | Qualitative data to support quantitative results would have been informative |
| 54. Thomaz et al., 2020, Brazil | Communication choices and strategies | Other—qualitative, semi-structured interviews | 10 caregivers of deaf children aged 10–19 years, special school in Southern Brazil | To understand the family interaction with the hearing-impaired child/adolescent | Identified that interaction of the deaf with the family and society is impaired by people's lack of knowledge about the deaf community and the Brazilian Sign Language, which raises concern in caregivers who often overprotect the child/adolescent which may limit the full development of their skills and autonomy | Study adds to discussion on theme and importance of sign language for communication with deaf people | Sample restricted to only one special school |

(*Continued*)

**Table 3.** (Continued)

| Study, year, location | Theme | Study design | Sample population | Study aim | Findings | Strengths | Weaknesses |
|---|---|---|---|---|---|---|---|
| 55. Trahan, 2016, USA<br><br>PhD thesis | Family perspectives and environments | Other—qualitative<br><br>Phenomenological study | Six parents (three Deaf, three hearing) of deaf children in Southwest USA | Examined positive and negative experiences faced by parents of deaf children going through the Individualised Education Plan process. | Parents do not have a true voice in the process because they do not understand the IEP and how they can advocate for their children, regardless of their educational levels or experiences. Complexity of procedural safeguards, parents' insufficient knowledge of IEP procedures, and lack of access to school personnel with fluency in ASL were identified as the barriers parents in this study experienced | Large number of schools contacted, participants from diverse cultural backgrounds sought to avoid identification and focus on any one state. Piloted interview questions on one deaf and one hearing parent. | Small number of participants, two were interviewed in Spanish by Spanish speaking hired researchers, possibility of dialect differences between participants and interpreters |
| 56. Watkins et al., 1998, USA | Interventions and resources | Comparative<br><br>Two groups: children had a deaf mentor do regular home visits (Utah) v children who did not (Tennessee). Both groups had visits from a parent advisor | Families in Parent Infant program under Utah School for the Deaf. 18 children in each group, matched by age and hearing loss | To compare children who received deaf mentor services to matched children who did not receive these services but who received parent adviser services | Children receiving this early bilingual-bicultural programming made greater language gains during treatment time, had considerably larger vocabularies, and scored higher on measures of communication, language, and English syntax than the matched children. | Two comparable groups. Parents in the Utah Deaf Mentor Program became more comfortable using both ASL and complete signed English as the project progressed. The Utah parents reported different attitudes toward deafness, ASL. and Deaf culture than the Tennessee parents. | Small sample sizes, an absence of reliability and validity' measures on the instruments developed specifically for the study, the lack of videotaping in Tennessee (project constraints made this unfeasible), and the limited number of measures used. |

(*Continued*)

**Table 3.** (*Continued*)

| Study, year, location | Theme | Study design | Sample population | Study aim | Findings | Strengths | Weaknesses |
|---|---|---|---|---|---|---|---|
| 57. Wiggin et al., 2012, USA | Interventions and resources | Other—quantitative<br><br>Pre-school and home environments analysed using Language Environment Analysis (LENA) | Children with varying degrees of hearing loss enrolled on auditory-oral 6 week part-time program<br><br>Children aged 3–5 years who were dear or hard of hearing and had an English language level of 18mths-4 years were recruited from Denver area.<br><br>8 children were recruited for the summer pre-school program | This study investigates the amount of language available to children in the home environment and a summer preschool program<br>And impact of reduced educational programs over summer months | The children received more complex language in the pre-school environment than in the home environment, and therefore benefit from summer pre-school programs. Parents also benefit from parental education about language strategies in the home environment | The 3 hour pre-school experience provided stimulation comparable to a 10-16hour day for hearing children. Pre-school experience allowed the children who took part to double their access to spoken language and to double their conversational turns. Data provides a comparison of two language environments (home and school) | Difficult to demonstrate language change over six weeks, and standardised test is designed for annual use. |
| 58. Wong et al., 2018, Australia | Deaf dichotomies | Other—quantitative<br><br>Survey included an adapted version of the Looman Social Capital Scale (LSCS), the Family Empowerment Scale (FES), and questions on psychosocial outcomes. | Responses from 16 adolescents (aged 11–14 years) and 24 parents were received<br><br>Recruited from Longitudinal Outcomes of Children with Hearing Impairment (LOCHI) large cohort of DHH children in Australia | To explore the social capital of Australian adolescents who are deaf or hard of hearing (DHH) and their parents, and investigate its relationship with individual child or family characteristics, language, literacy, and psychosocial outcomes | On average, parent-rated social capital was positively related to adolescent-rated social capital.<br><br>Higher adolescent-reported social capital was reported in those with no additional disabilities. Aspects of adolescent-reported social capital were significantly related to their language and reading skills, but not with psychosocial outcomes | Current findings lend some support to promoting social capital in adolescents who are DHH and their families. The areas of social capital that were rated lower overall were bridging- or linking-related domains, such as working with other families like their own, and feeling a sense of power/control over community and political-level decisions regarding the child. | Small sample size and low response rate. This study also adapted the LSCS and FES that were originally developed as parent-report scales, for use in adolescents. The direct assessments were completed when the child was 9 years old and so collected between 2–5 years prior to the social capital data collected in the current study. Some of the children may have improved (or declined) in regard to their current language/ literacy abilities |

(*Continued*)

**Table 3.** (Continued)

| Study, year, location | Theme | Study design | Sample population | Study aim | Findings | Strengths | Weaknesses |
|---|---|---|---|---|---|---|---|
| 59. Wright et al., 2021, UK | Interventions and resources | Other—Literature review, eight databases were searched for early interventions for parents of deaf infants | N/A | To identify available literature for early parenting interventions for deaf infants, to synthesis targets and to highlight evidence gaps | Identified parent support interventions included both group and individual sessions in various settings (including online). They were led by a range of professionals and targeted various outcomes. Internationally there were only five randomised controlled trials. Other designs included non-randomised comparison groups, pre / post and other designs e.g. longitudinal, qualitative and case studies. Quality assessment showed few high quality studies with most having some concerns over risk of bias. | Prior to the scoping review national workshops and parent meetings were held with outcomes p informing the search strategy | There were concerns over risk of bias in many of the studies assessed |
| 60. Yoshinaga-Itano, 2003, USA | Interventions and resources | Other—quantitative<br><br>Overview of Colorado Home Intervention Program, (CHIP) universal newborn hearing screening programs were established in Colorado, changing the age of identification of hearing loss and initiation into intervention in this program geared to families with | Families with children from birth to 3 years | Report on a series of studies, from 1994 to the present, investigated predictors of successful developmental outcomes. The article provides information about how the findings of these studies relate to the existing literature. | Language development is positively and significantly affected by the age of identification of the hearing loss and age of initiation into intervention services. Both speech development and social-emotional variables are highly related to language development | Results replicated in Nebraska (Moeller) and Washington (Calderon). This study includes longitudinal and cross-sectional studies. | Limited details of results of different studies over past nine years as this is an overview paper |

(*Continued*)

**Table 3.** (Continued)

| Study, year, location | Theme | Study design | Sample population | Study aim | Findings | Strengths | Weaknesses |
|---|---|---|---|---|---|---|---|
| 61. Young et al., 2006, UK | Communication choices and strategies | Other—Literature review, term 'informed choice' searched over 10 databases, 927 hits, reduced to 152 selected articles | N/A | A theoretical discussion of the problems associated with the concept of informed choice and deaf child services and then focuses specifically on why a meta-study approach was employed to address both the over-contextualized debate about informed choice | Overarching themes: a) the nature of information an b) parameters and definitions of choice. Research team reported integrated understanding as focus less on informed choice and more on implications of informed choice interacting with personal and structural circumstances | Insights and debates for research team to broaden understanding and to critically reflect on researcher assumptions at start of informed choice guidelines project | Studies identified not listed |
| 62. Young et al., 2008, UK  Lit review | Deaf dichotomies | Other—Literature review  4 databases searched, and 130 full text articles reviewed | N/A | To consider insights and experience of resilience research generally might be applicable to, or modified by, the specific conditions of working in deafness | Evaluated the implications of what little deaf-related resilience work exists for future directions for research and, ultimately, the promotion of resilience-enabling interventions in this field | | Researchers note limited literature available |
| 63. Young et al., 2005, UK [122] | Communication choices and strategies | Other—qualitative, focus groups | 20 Deaf and hearing parents participated in four focus groups. Parents recruited from a National Deaf Children's Society database | To review a standard information folder for parents of newly diagnosed deaf children that was being developed by the National Deaf Children's Society and government-sponsored Early Support Pilot Program. | Parents take a meta-analytic role about the processes of information provision, not just a micro-analytic role in the evaluation of what is produced. Second, many of those difficulties pertinent to the provision of information for parents of deaf children clearly have wider applicability | Research and development collaboration between government, research and voluntary organisations ensured development of a quality information product | Unable to gather parental experiences to add, due to time constraints |
| 64. Young, 1999, UK | Deaf dichotomies | Other—qualitative  Interviews  Families had visits from Deaf consultant (Deaf role model), Teacher of the Deaf visits, and parents and children attending pre-school nursery group | 12 parent/carers (9 mothers, 2 fathers, 1 grandmother—of nine severely/profoundly deaf children overall aged 20–32 months), six hearing teachers of the deaf, six Deaf consultants | To explore the process of adjustment to having a deaf child. | Two key themes: parents' search for their child's world, relationship between childness and deafness  The cultural-linguistic model of deafness raises new adjustment issues for parents of deaf children, as hearingness needs to be explored as much as d/Deafness | Early intervention program had been running for three years | Small sample |

(*Continued*)

**Table 3.** (Continued)

| Study, year, location | Theme | Study design | Sample population | Study aim | Findings | Strengths | Weaknesses |
|---|---|---|---|---|---|---|---|
| 65. Zaidman-Zait, 2007, Canada | Family perspectives and environments | Other—qualitative<br><br>Interviews—critical incident technique to identify significant behaviours, thoughts, feelings that facilitated parenting experience | 15 hearing mothers and 13 hearing fathers (12 married couples) whose children had cochlear implants | To describe and categorize the attributes that parents of young children with cochlear implants (CIs) consider as facilitating their parental coping experience | 430 critical incidents were identified and sorted into 20 categories<br><br>The current research substantiates the soundness of implementing early intervention models such as the developmental system model (Guralnick, 2001 [120]) and the support approach to early intervention (McWilliam & Scott, 2001 [121]), which coincide with ecological theory and recognize that families need various combinations of resources, social support, information, and services to help them address the stressors associated with parenting in general and parenting a child with special needs in particular. | Findings supported by other studies (Sach & Whynes) | Small sample size |

from sources, a thematic framework was applied to categorise the areas of support for hearing parents. This involved sorting studies into categories as follows: i) *Communication choices and strategies; ii) Interventions and resources; iii) Family perspectives and environment; and iv) Deaf identity development*. In addition, strengths and limitations of the sources are presented in Table 3. Context from the grey literature is included in this paper's introduction section, as this clinical wisdom provides additional information and context.

# Findings

## Theme one: Communication choices and strategies

Hearing parents will need to decide whether their deaf child will communicate using a spoken language or a signed and spoken language [37]. The timing of this communication choice is challenging as hearing parents make decisions during the small window when their child starts to develop language during the first few years of life. Hearing parents have little understanding about deafness, nor is infrastructure present to guide parents towards appropriate engagement with Deaf communities to begin discussing the differences between communication strategies. Parents can be inundated with information regarding communication and educational methods [20]. Yet the decision is up to parents and the key factor being that any form of early

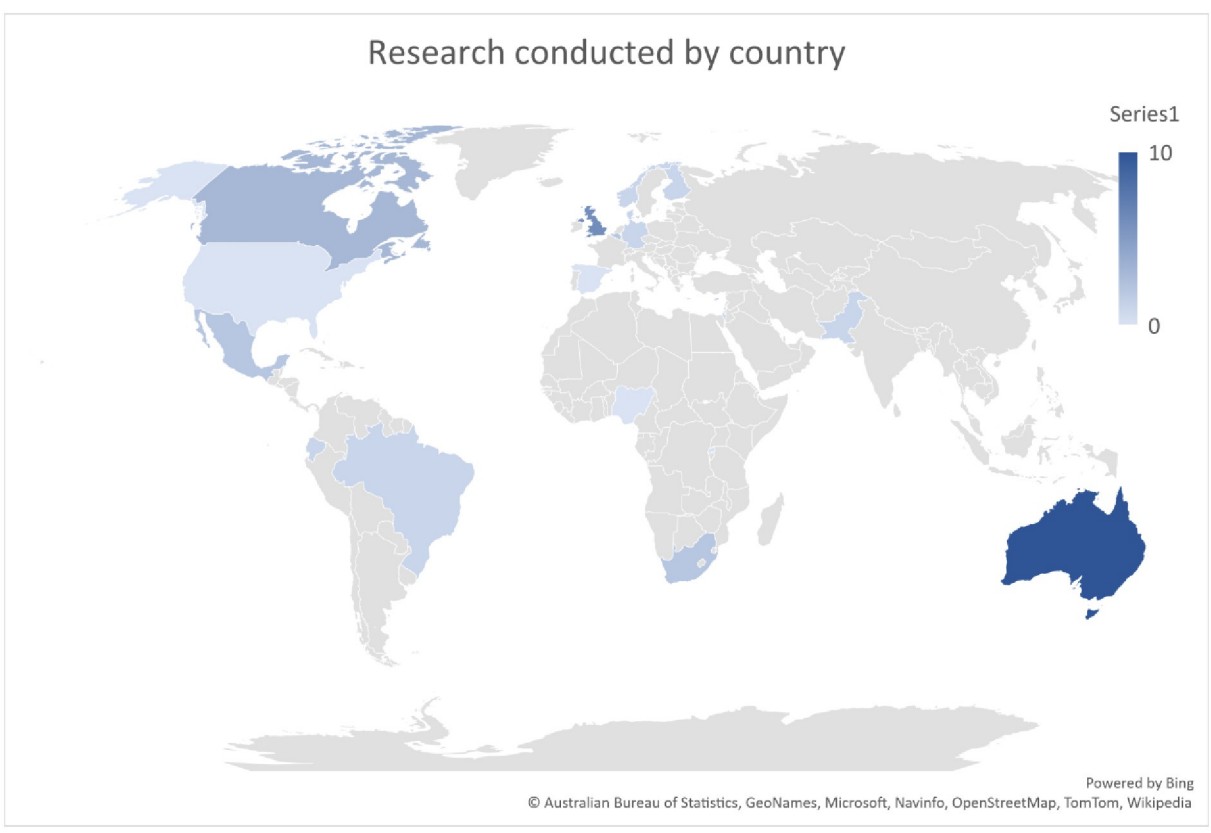

**Fig 2. Research conducted by country.**

language development is critical [19]. Around the decision-making time, parents commonly want to know what will give their deaf child the best chance of learning to communicate, and whether using sign language might adversely affect their academic achievements and if it is worth waiting to see the impact of a cochlear implant before learning sign language [5]. There is frequent reporting that medical professionals claim that promoting a signed language with a deaf child may delay or hinder the development of spoken language learning, with suggestions that children may be confused [5], although much evidence supports the positives of learning to sign [19].

Retrieved papers under the communication choices and strategies theme included 20 primary research studies and two literature reviews. The 20 primary research studies included three co-comparative studies, four quantitative studies and twelve qualitative studies and one PhD thesis.

Factors contributing to parents' selection of a communication mode to use with their children with hearing loss, are reported as information, perception of assistive technology, professionals' attitudes and the quality and availability of support [38]. Parents' decisions about communication choices with their deaf child are strongly influenced by the information they receive, which in the main focuses on amplification of sound, with information givers rarely mentioning sign language approaches [39]. Parents who chose speech only as a communication choice appear to have received advice from education and speech/audiology professionals more often [37]. Similar findings are reported in other studies that parents relied heavily on advice from professionals [40, 41]. There is suggestion that advice from speech and language

professionals, audiologists and specialist teachers was valued by parents over medical or non-professional views [42]. Conversely, parents of deaf children they surveyed did not find any professional group's advice more influential than another, and reported they ultimately relied on their own judgements to make decisions about their child's communication choices [43].

Several studies in this scoping literature review compared hearing and Deaf parents' views about communication choices as well as child outcomes. Deaf parents are likely to choose a more visual mode of communication for their deaf child, and frequently outperform hearing parents in interaction studies that compare hearing and Deaf parents' engagements with their deaf children [44]. For example, Deaf parents tend to use a higher level of tactile strategies when communicating with their deaf child compared to hearing parents [42].

When parents make hearing technology and communication modality choices for their children amongst competing discourses of deafness and language, hearing program principles of fully informed choice of communication narrowly reflected medical knowledge of deafness only [43]. Frequently there is reported to be minimal information about sign language and Deaf culture, and over time parents resist medical knowledge and asked for alternate services as their knowledge of their own children grew beyond diagnostic assumptions [43]. Initial adoption of a medicalised model script is recognised as occurring, which often maintains a strict divide between competing views of deafness [44], such views may include parents thinking their children are successful if they do not need a signed language.

In a comparative study two groups of hearing mothers with deaf children were studied, with one group more experienced as their children had been diagnosed for more than 24 months (compared to the mothers with children diagnosed in the last 18 months) [45]. The aim was to investigate the type of communication strategies that parents use with their children and how the type of early intervention (EI) involvement affected parents' values about communication strategies. Mothers completed questionnaires about their views on communication strategies and were also videoed for 3-minute mother-child play interactions, and only minor differences were found between the groups of less and more experienced mothers of deaf children suggesting limited impact of early intervention programs on parental choice of communication method [45].

The main factors that influenced caregivers to change the communication method with their child with hearing loss included family characteristics, access to information [46], family strengths, family beliefs, and family practices, with the family at the core of decision-making regardless of severity of hearing loss, family demographic or type of device used or communication approach [47]. Similarly, the importance of communication changes regarding language modality being child-led, as parents adapted their language choices in line with their child's needs to improve communication confidence, noting that early sign language exposure benefits the development of spoken language [48].

A comparative study of hearing versus Deaf parents with their respective deaf children acknowledged the active role that parents, and children take when communicating as they sought to explore successful joint attention (where one party seeks to gain the attention of the other, and the other responds) [49]. Studies that inform understanding of factors supporting language success are crucial. Communication in families will always be a joint venture and knowing if gaining joint attention at an early development point would assist families and professionals with communication choices in the future [49]. Very often parents want to know exactly what it is that will help communication to be most effective.

Complexities of communication choice are apparent in studies that focus on the intricacies of self-identify in children of parents who chose sign language as a primary mode of communication [50]. Follow up appointments focusing on communication modality, particularly following cochlear implantation, suggest a background of opposing views on communication

choice mean increased awareness for parents is vital [51]. Families can unknowingly overprotect their child, limit knowledge and skill development due to hearing parents' lack of knowledge and understanding about Deaf culture and Deaf communities [52]. All three studies highlighted the importance of continuing professional development for workers in order that they gain familiarity with these topics, and in turn discuss them with families of deaf children [50–52].

Perceptions of factors that foster success in deaf students from parents, teachers, interpreters, notetakers and deaf students themselves do not mention communication choice at a young age; instead, success was attributed to strategic components including self-determination, family involvement, friendships, reading and high expectations [53].

In one study deaf children of Spanish-speaking families studied did not learn American Sign Language (ASL) early on, often coming to this much later, with many of the children having limited access to language early on, and parents expressing frustration at not being able to communicate with their children, with the family being left behind through delaying communication through ASL [54].

The importance of professional advice provided to hearing parents of deaf children about communication mode and language use choices is noted, as this may heavily influence caregiver choices about communication. Understandings about factors that led to specific communication choices by hearing parents could be gained through further research [55]. The next theme focuses on papers concerning interventions and resources that support hearing parents with deaf children.

## Theme two—Interventions and resources that support hearing parents with deaf children

Theme two incorporates identified studies that focused on interventions and resources that support hearing parents with deaf children. In this section we report on intervention programmes for hearing parents with deaf children broadly, then how programs were delivered and finally specific types of interventions that support hearing parents with deaf children.

**Specific interventions of Deaf mentors and role models.** A scoping review of early interventions for parents of deaf infants [56] found that interventions commonly focus on language, communication and parent knowledge, well-being and parent/child relationships and did not find any studies focusing on parent support to nurture socio-emotional development, which is often a poor outcome for deaf children. Socio-emotional development is not well-analysed by hearing professionals, who may not realise that it is not deafness that needs fixing but everything around it. It was concluded that research in this area is much needed, with most studies conducted some time ago and not in line with healthcare advances, recommending further research to develop evidence based early intervention [56]. A literature review of early intervention programme models and processes [57] identified five themes which were caregiver involvement, caregiver coaching, caregiver satisfaction, intervention program challenges and telehealth. Understandably caregiver involvement needs to be culturally and linguistically appropriate, as this improves caregiver satisfaction with services and improves outcomes for deaf children [57]. Another example is the HI-HOPES intervention program, developed in 2006 and still current, with an appreciation of South Africa's characteristic linguistic, racial, and cultural diversity, noting embedding of cultural values and practices and includes provision of Deaf mentors [58].

A series of studies of the Colorado Home Intervention Program over nine years [59], saw a change in the average age of intervention decrease from 20 months to 2 months, meaning infants had much earlier intervention and therefore increased their language and social-

emotional range. The early engagement with parents from a CO-Hear co-ordinator about choice of intervention service is a key success factor [59]. Another language intervention program with a sole parent focus, this time oral only, is the Muenstar Parental Program [60], a family-centred intervention following newborn hearing screening. Parents received training on the positive impact their behaviour had on their infant including showing more eye contact, more imitations and more listening, where parent and trainer discuss and agree principles to intensify in the next videotaped interaction. Although only single training sessions [60], authors noted the model to be a comprehensive early intervention focusing on encouragement, however, when published it was at the concept stage with minimal data available.

Summer pre-school language environments compared to their home environments suggest there are benefits to children, whilst recognising that pre-schoolers' parents continue to require education around language strategies [61]. Parents would likely benefit from guided practice regarding extending conversations and asking questions at their child's language level, and how to expand their children's language, and that practising these skills with a professional is essential [61].

Mentors for families with deaf and hard of hearing children have been found to be highly effective, with study examples of family mentors [62] and mentors for children [63]. There is an awareness that parent- to- parent support models are rooted in disability ideologies and are highly valued [64], and often need to be unique [62]. Parent mentors made notes following each phone support conversation, and notes analysed over a two-year period showing hearing related conversations, early intervention and multiple disabilities were the primary topics of conversations between parent mentors and families. A literature review and eDelphi study to define the vital contribution of parents in early hearing detection and intervention programs suggested supporting, or a mentoring parent was well received [65].

Similarly Deaf adults are a key element in early intervention programs [66], primarily as role models and language providers, noting that families do not have a range of Deaf professionals to connect with in early intervention programs. One of the first reported studies of Deaf family mentors [63] provided a Deaf adult mentor who made home visits to deaf children and their families to share language, as well as a hearing advisor to support to parents. This type of provision is referred to as bilingual-bicultural and was intended as introductory in the first instance in two US states. The Deaf mentors taught each family American Sign Language (ASL) signs, interacted with the child using ASL, shared Deaf knowledge and culture and introduced the family to the local Deaf community, promoting a bi-bi home environment. It is reported children with Deaf mentors used more than twice the number of signs and parents used more than six times the number of signs than the control group [63]. 85% of survey respondents in the Lifetrack Deaf mentor family program operating in Minnesota USA reported their child's quality of life to have improved, and 76% of families finding the information about Deaf culture 'very helpful' [67]. There are limited examples of early intervention providers that include Deaf mentor provision for children and families in the US, and whilst 27% of their survey respondents said there were a diverse range of Deaf professionals for families to connect with; but only 2% of respondents reported the first point of contact with early intervention professionals had been with a Deaf person [66].

**Delivering intervention programs using telehealth.** Although the provision of healthcare with remote support has become commonplace during Covid-19, prior to the pandemic many services used telehealth because it offered the potential to meet the needs of underserved populations in remote regions [68, 69].

Tele-practice or tele-intervention (or virtual home visit) has been used increasingly as a method of delivering early intervention services to families of deaf children. Tele-practice intervention outcomes were compared for children, family and provider compared to in-

person home visits using fifteen providers across five US states (Maine, Missouri, Utah, Washington and Oregon) and found children in the telepractice intervention group scored significantly higher on their receptive and total language scores that the children who received in-person visits [70]. Higher scores were also reported with telepractice intervention for parent engagement and provider responsiveness compared to in-person visits [70]. Parents reported having better support systems, feeling better supported by programs and knowing how to advocate more for their deaf child. Notably in-person visits were reported to focus more on intervention with the child with parent observation, whilst tele-practice engaged parents more in supporting parents as the child's natural teacher. Equally when comparing tele-intervention with in-person visits, increased engagement from the tele-intervention group has been reported, with families reporting themselves to being 'more in the driving seat', and specialised early intervention services for families with deaf children via telehealth to be cost effective [71].

Preschool and school services were examined for children who are hard of hearing and described service setting, amount, and configuration, analysing relationships between services and hearing levels and language scores [72]. Noting that as children reach the age of three years that services often shift from being family centred to being more child focused and a need for more interprofessional practice to best meet the needs of children who are deaf. Findings that 19% of families did not receive any intervention, which rose to 30% by the time children were of school-age [72].

**Intervention support—Teaching sign language to parents.** Another specific type of intervention to support hearing parents with deaf children is supporting the teaching of sign language. When deaf children are introduced to sign language there is an obvious need for parents and significant others in the child's situation to learn to communicate in that language. However, if there are no other Deaf members of the family, a signed language may not be used in the home. Therefore, the deaf child may not have the exposure to language role models in the home in order to acquire a signed language as a first language. Giving parents a way to communicate with their deaf child will mean parents are provided with greater opportunities to engage effectively with their child's world. It may be that a signed language does indeed later become a deaf child's primary language, and early development of this in the home can be key. Six key components in any language development and support programme for parents include communication strategies, language tuition, immersion/language use, language modelling, information giving and practical/emotional support [73]. During curriculum development of Australian Sign Language (Auslan) and creation of family-specific resources, after finding the need for a language development program that incorporated classroom teaching, incidental learning opportunities and natural sign language immersion with additional learning resources. There is limited available evidence on the teaching of a signed language but researchers stress the need for involvement with Deaf adults or what it is like to live as a Deaf person being of primary importance [73].

A five-year sign language intervention project is reported [74] with 81 hearing family members in Finland learning sign language once a week with a teacher who was Deaf, Parents, siblings, and other relatives met once monthly to study sign language, and all families in the project signed together about twice yearly. Noting that if one is to succeed in modern society, communication competence should be good [74], and found that families most actively involved learned a greater amount.

One challenge noted by research teams regarding interventions given the geographic dispersion of children who are DHH is the shortage of adequately trained professionals [70, 71]. The next theme presents material from the literature about family perspectives and environments.

## Theme three: Family perspectives and environments

Family perspectives and environments are an over-arching theme that include evidence about family experiences, needs, coping and environmental relevance, and are reported in this section.

A study of family experiences and journeys exploring reactions, behaviours and strategies with 50 hearing parents with deaf children in Karachi City, Pakistan [75], found all parents reported shock on learning their child was deaf, and 99% were stressed by this news. 98% of these parents wanted counselling and support about three main areas: diagnosis of hearing impairment, speech and communication, and hearing aid maintenance, with specific structured counselling and information sessions in hospitals or schools recommended [75]. Family journeys with childhood deafness in Mexico are explored through the lens of a pilgrimage through Pfister's [76] study as families realised their quest was not about fixing hearing but about finding more reliable communication methods. Parents reported the most common support was in the form of biomedical options which had restricted scope. Families also reported countless troubling questions without a forum to present them [76]. Similar to the concept of impairment as a predicament that can be overcome [77], families wanted to continue their quest for worlds people inhabit and aspired to challenge medicalised ideologies, which suggest family perseverance [76]. Eighteen hearing parents of deaf children in Western Australia reported struggling with a deafness diagnosis and recommendations for professionals who should not "just give a pamphlet to parents. . .never assume technology will cure all. . .and find out how a family ticks" [78]. It is stressed that more research is needed about deaf children with hearing parents across various life stages to fully understand potential challenges; and concluded that Deaf parents of deaf children have much insight to offer hearing parents with deaf children.

Studies that examined hearing families' stresses and needs highlighted socioeconomic and cultural factors impacting on carers of deaf children in Ecuador around education and employment [79]. Carers are critical of new measures around schooling that may lead to reduced resources and discrimination and propose future healthcare practitioners screen deaf children for potential abuse regularly due to their vulnerabilities. Using the Parenting Stress Index and information gathered on personal and social resources, researchers found parent variables are largely responsible for successful child development [80]. A correlational study of stress levels and coping responses found the relationship between family and parental stress and a crisis with a child with a disability to be complex [81]. Notably families who were able to communicate with their deaf child through a signed language found this was positively related to their stress experience.

Parenting stress reported by Korean mothers of deaf children [82] suggests a need for comprehensive support services that include schools, parents, siblings and social workers, as they reported on-going alienation in mainstream education [83] and feeling left out within family relationships. Having a child with hearing loss does change family dynamics as hearing loss becomes the dominant family topic. Healthy families of children who were deaf were interviewed to identify what contributed to a health family dynamic [84]. Finding that families engage with a variety of professionals, there was a reported desire for professionals to more actively listen and to demonstrate confidence in families to capitalise on existing strengths and resources [84]. Proactive families welcomed workers who were willing to tolerate a variety of perspectives and options for them and their deaf children, and for workers to create social events for families and workers to interact together. Often hearing families report not having a true voice because they do not understand educational processes and systems, which does not help them to advocate for their deaf children [85].

Researchers who explored coping strategies of parents with deaf children note that parent stress is not an outcome of child deafness but of different characteristics of the context, perceptions and resources [86]. Exploring critical incidents with parents whose children have Cochlear implants to understand what influences parents' coping suggest opportunities to share experiences with others and consistent family support are essential, as is the importance of understanding what hinders coping processes [87]. Adolescents themselves with Cochlear implants in Copenhagen reported diverse experiences from others of similar age, with participants reporting higher levels of feeling different from others also reported higher levels of loneliness, although this was less for those implanted at a much earlier age; and implies the need for flexible tailored support for all [88]. The actual reasons for deaf adolescents reporting loneliness is not fully known. Family environments can be enhanced by education and therapy to create robust language environments to maximise cochlear implanted children's potential [89]. Families who reported they had a higher emphasis on being organised self-reported they had children with fewer inhibition problems, and that emphasis on structure and planning in family activities can help grow a supportive social family climate. Family environments are one area that can be modified when families become aware of problems impacting on their child's progress [89].

A historical study conducted in Cyprus reported on Deaf adults' childhood memories and how when they were children they reported feeling isolated in family environments due to lack of communication as families often refused to learn sign language [90]. This worsened when extended family visited and speech pace increased. The 'dinner table syndrome' is much reported and describes indirect family communication that occurs at family meals, during recreation and car rides that provides important opportunities to learn about health-related topics and are common to most families [91]. Deaf people with hearing parents often report limited access to contextual learning opportunities during childhood [92] which highlights the importance of environmental factors.

However for deaf children introduced to Deaf adults in Deaf clubs there are clear benefits for engagement with Deaf role models, where they can discuss serious issues and communicate effectively [76, 90]. Although it must be noted that Deaf clubs in many parts of the UK and US are reducing in number [93, 94]. Social success can be viewed differently, with hearing children and parents seeing their friendships more positively than deaf children [95]. Evidence is consistent about deaf children with Deaf parents having higher social success and better communication outcomes than deaf children of hearing parents [95].

## Theme four: Deaf identity development

Deaf identity development describes the contrasting nature of opposing aspects of deaf and hearing perspectives on topics that relate to support for hearing parents, for example models of deafness and language and communication modalities, as well as ways deaf people encounter Deaf identity.

A review of mainstream resilience literature, in relation to what it means to be deaf and the contexts of deafness around disability, suggest resilience is often about challenging social and structural barriers [96]. The barriers in themselves often create risk and adversity, and for deaf young people the successful navigation of "countless daily hassles, which may commonly deny, disable or exclude them" is a key definition of resilience [96, p 52]. Protective factors and skill development are the enablers.

The cultural constructs of deafness and hearingness can best be viewed through a lens of multimodality, with communication being more than about language [97, 98]. The focus on why the body matters in how we, hearing and Deaf, come to shape a sense of self and the

interplay between resources we use in the process. Such intersections are important in the development of identity and social skills. Aspects of adolescent-reported social capital (for example, the networks and relationships that enable a society to function) are reported as being linked to their language and reading skills, with deaf young people found to have less strong social skills than their hearing peers [99]. Aspects of adolescent-reported social capital are positively related to their language and literacy outcomes, suggesting the importance of increased promotion of social capital in adolescents who are DHH and their families [100].

The importance of understanding different ways that deaf children are contextualised, usually through the medical model, the social model and the Deaf culture model of Deafness are reported [101], with the medical model remaining dominant and framing being deaf as having hearing that does not work and needs to be treated to restore Deaf people to the normality of the majority of the population. The social model of deafness focuses on disability and strives for inclusion to ensure differences are supported. A Deaf cultural model values and celebrates Deafness collectively, often with a focus on Deafhood [102] and Deaf pride, where the label of impairment is seen negatively. A social relational model that is more about how deaf children shape their own identities and relocating the balance of power to create policy directives regarding increased use of signed language would enable greater inclusion and would directly challenge structures that exist [101]. Re-framing deaf children as plurilingual learners of signed language, English and additional languages, instead of as deficient bilinguals by dominant culture standards has potential [101].

Hearing parents' experiences of adjusting to parenting a deaf child is impacted by the cultural-linguistic model of deafness have been examined, and how challenging the notion of a loss or deficit and instead using a model which promotes a linguistically able and culturally diverse lens [103]. An early intervention programme in the UK involving hearing parents and hearing teachers where families received weekly visits involving Deaf consultants in the role of 'Deaf friend' engaged family members in games, discussion and sign language tuition. Two key findings were reported with parent anxiety about the meaning of deafness reported as lessened by a Deaf adult 'simply being themselves' [103, p163]. Equally, the relationship between childness and deafness, concerned with the overlap of a child being both a child, and a deaf child, and the importance of accepting the child and their child's deafness. The cultural-linguistic model of deafness on the adjustment process hearing parents of deaf children experience is a potential tool to support parents through their reactions to their child's deafness [103].

The discursive context of cultural-linguistic model views and medical models of deafness perspectives is present in hearing mothers' talk and how they positioned their meanings of the two phenomena [104]. The language of advice from professionals has substantial influence, and positioning theory helps to explain the discrepancies parents experience between reported and actual plans for language practices [104].

An in-depth analysis of a shared-signing Bedouin community [105] highlights how deafness does not easily fall under the medical model because a wider lens is used in communities where many individuals who are hearing sign too, similar to Martha's Vineyard situations [106]. Evidence is generally about Deaf communities rather than signing communities [106], and how linguistic communities do not just share a language but knowledge of its patterns of use and its cultural distinctions (such as attention getting and name giving) can be key in terms of identity development.

Descriptions of the Deaf Bi-lingual Bi-cultural community (Bi-Bi) helps us to understand this unique identify in an increasingly diverse world, and the relationship between language and identity formation and people's social participation [107]. Misconceptions about bi- and multilingualism frequently recommend families limit their deaf child to learning oral English

only, although multiple languages result in fluid conversational exchanges, trusting parent relationships and a strong cultural identity. Increasing clinicians' understanding of language and culture, particularly Deaf culture would mean they could more effectively support child development and respond to human diversity issues in healthcare environments [107].

The importance of signed stories and how Deaf teachers' storytelling in schools is an important part of deaf children's identity development [108, 109]. Due to the decades of strict oralist policies (from 1880 to 1980) [110], many deaf children do not experience the possibilities of a Deaf identity unless they go to a deaf school due to the lack of employment of Deaf teachers in mainstream education. Signed stories are a way of teaching deaf children about their linguistic and cultural heritage [108]. Rather than conceptualising deaf people as individuals who cannot hear, Deaf people see themselves as viewing the world visually and often use sign language, so deafness is not a loss but a social, cultural, and linguistic identity.

## Discussion

The aim of this scoping review was to identify published evidence on the supports and structures surrounding hearing parents with deaf children. The characteristics and results of the included articles were assessed. To the authors' knowledge this is the first scoping review that focuses on what supports hearing parents as they in turn nurture their growing deaf children. Following a thorough database search and eligibility criteria, 65 papers were included in this scoping review. While it is a large amount of evidence about what supports hearing parents with deaf children, the evidence is mainly based on small, non-repeated studies with few randomised controlled trials published on the efficacy of support for families with deaf children. Current knowledge has therefore been framed as a narrative synthesis of reports of what supports families.

When families with deaf children are introduced to communication choices and strategies, their decisions are strongly influenced by the information they receive [46], but ultimately, they rely on their own judgements, with family characteristics, family strengths and beliefs also considered [47]. Hearing parents are less likely to choose a visual mode of communication, which may be due to hearing programme principles reflecting a predominantly medical model of deafness resulting in more ableist and audist approaches [43], although some parents do go on to ask for alternate services over time as their own knowledge of their child grows. It is reported that there are three phases of decision-making—information exchange, deliberation, and implementation, with two key decisions dominating on implantable devices and communication modality [111].

When discussing communication choices with families, there is a need for professionals to be familiar with and understand the cultural ecology [12, 46] and that parents may make choices without access to information, and that not all choices are available. Culturally incompetent care often spreads health inequalities for Deaf people [28]. Increased awareness of communication choices is vital for parents because families may unknowingly limit knowledge or skill development due to limited awareness of Deaf culture and Deaf communities [52].

Studies that were categorised as providing evidence about interventions and resources that support hearing parents made mention of the value of interventions that focused on language, communication and parent knowledge as well as supporting parent-child relationships. There was a paucity of evidence about nurturing socio-emotional development which is often a poorer outcome for deaf children when compared to hearing children [56]. There was an emphasis that intervention programmes need to be culturally and linguistically appropriate, as this improves caregiver satisfaction [57], and that all interventions with families need to address linguistic, racial and cultural diversity elements.

The provision of Deaf mentors was noted to be a popular feature with families [58–60]. Although there are often few Deaf professionals in services for families to connect with and limited evidence of sustained Deaf mentor programmes available [66]. A supporting parent was also a welcome intervention, which carried less sense of a hierarchical relationship and families reported valuing such input [65].

There is evidence to suggest that intervention and support occurring early result in better language for deaf children at later point [59]. Giving parents guided practice with examples for their individual child's language level and practice of this skill with a professional was highlighted as useful [61]. Increasing evidence suggests that deaf children having access to a signed language at the earliest possible age is beneficial [22] but it must be noted that Deaf people's under-achievement in education is not a result of deficits within children themselves but relates to the 'disabling pedagogy' to which they are routinely subjected [112].

Whilst many services have moved online during the pandemic, the reported results for parent intervention with deaf children are before Covid-19 occurred, with telepractice groups scoring significantly higher on their total language score and more in the 'driving seat' [73] which may be due to parents saying they felt better supported and engaged through this route. As deaf children grow older, and services move to being more child-focused than family-focused there is evidence that families voice feeling less supported with over 30% reporting no intervention by the time children attend school [76].

The reported key components of language and support programmes for parents are that communication strategies, language tuition immersion and language modelling, as well as information and emotional support are all essential [73]. It is not uncommon for support programmes to include family get togethers sporadically, say two to three times per year [74].

The family perspectives and environment theme included reports that 98% of hearing parents wanted counselling on discovering their child was deaf [75]. A priority for parents was finding reliable communication methods, and whilst parents had commonly been offered biomedical options and information, many suggested they wanted a forum to raise concerns and questions [76] and did not want to overly rely on medicalised ideologies [77]. More information was wanted from hearing parents about challenges they might encounter at different life stages for their child [78].

Environments for deaf children need vital consideration due to the potential for abuse of vulnerable groups [79]. However, parent variables are largely responsible for successful child development [84]. One example being parents who were able to communicate through sign language found this significantly lowered their stress as communication with their child was available to them [81].

Families were keen for professionals to value their strengths and resources, and particularly for social events to be arranged with other families with deaf children [84]. Parent stress seems to be more related to context and resources than actual child deafness [86] and knowing what hinders coping would be useful knowledge [87]. Enhancing family environments with education and therapy or therapeutic support is key [89]. Environmental factors for hearing parents with deaf children are vital, which is particularly evident with discussion of the dinner table syndrome with children missing out on many learning opportunities and family relational communication [91]. It is notable that deaf children with Deaf parents frequently outperform deaf children with hearing parents because of their early language encounters and immersion in an inclusive world [95]. The reverse is true for deaf children.

The theme Deaf identity development highlighted the importance of the intersectionality of Deaf identity in relation to other cultural identities [99]. Successful identify development is strongly linked to social capital [100], so rather than being contextualised by the social model, the Deaf culture model of deafness offers a more positive view which may empower both

hearing parents as well as their deaf children [102], as this challenges a deficit model and promotes a more linguistically able and culturally diverse lens [12]. Tools that promote acceptance of deafness, adjustment and managing reactions have much scope [103].

The language of diagnosing and medical professionals can have substantial influence, as well at the position that they take [16]. Communities that include hearing signers have much to offer, as the notion of signing communities suggests the benefits and richness of signed languages [106]. It is worth noting that most deaf children are not exposed to the idea of a Deaf identity unless they go to a Deaf school and have exposure to deaf children and Deaf adults on a regular basis. Since the evidence search for this scoping review was undertaken, further publications also support our conclusions. Namely that health care professionals and early intervention providers must inform parents about signed language as a language choice as the majority of parents only learn about such options through their own research [Lieberman]. Also that supporting parents' development of communicative competence in signed languages has significant implications for meeting their deaf children's communicative needs [112, 113].

## Limitations

A systematic and rigorous approach was adopted when carrying out this scoping review. Evaluating the findings of this scoping review the limitations are discussed in this section. The inclusion criteria were purposively broad at the outset, and due to the high number of retrievals it became clear that focusing on empirical research studies would provide the most valuable evidence. However, whilst some support programmes had been sustained over time, many were short term projects with small samples.

One limitation could be that only articles published in the English language were included in the review, therefore articles in other languages may have been missed in the search. Support systems for hearing parents with deaf children vary greatly. A formal quality appraisal of the included articles was beyond the scope of this review. To decrease the risk of bias the selection of retrieved papers was monitored and viewed independently by two researchers with differences of opinion resolved through discussion. A total of four electronic databases were selected and searched, and despite those covering a range of academic fields their databases may potentially have been excluded. However, at the outset the suggestions of 50 keywords/terms from the steering group helped ensure that a diverse and broad range of material was included. One limitation of this scoping review is that results are presented in a narrative style with limited quantitative analysis of retrieved studies. Whilst sample size and results are available in Table 3, there was a low number of randomised controlled trials on this subject and suggests that the evidence is available about what supports hearing parents with deaf children are not adequately addressed. Despite these limitations this scoping review provides what we believe to be a first overview of existing research on supportive interventions and help for hearing parents with deaf children and serves to highlight the lack of evidence on this important topic.

## Conclusion

Overall, the results of this scoping review about supports for hearing parents with deaf children suggest it is important to identify the journey parents and their children navigate from the results of hearing screening or deafness diagnosis, through to the available provision and supports from various services and providers. The results suggest that more research is needed to know what supports hearing parents with deaf children. We propose that further longitudinal studies should test and compare specific interventions and programmes in low-middle income countries and high-income countries. This scoping review highlights a need for improvement

in the experience of hearing parents with deaf children as they, along with their deaf children, navigate challenges, information provision and supports required.

## Supporting information

**S1 Checklist. The Preferred Reporting Items for Systematic reviews and Meta-Analyses extension for Scoping Reviews (PRISMA-ScR) checklist was used during the writing of this paper (see additional file PRISMA-ScR-Fillable-Checklist_SUPERSTAR 280623.tif).**
(TIF)

## Author Contributions

**Conceptualization:** Julia Terry.

**Data curation:** Julia Terry.

**Formal analysis:** Julia Terry.

**Funding acquisition:** Julia Terry.

**Investigation:** Julia Terry.

**Methodology:** Julia Terry.

**Project administration:** Julia Terry.

**Resources:** Julia Terry.

**Software:** Julia Terry.

**Supervision:** Jaynie Rance.

**Visualization:** Julia Terry.

**Writing – original draft:** Julia Terry.

**Writing – review & editing:** Julia Terry, Jaynie Rance.

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
