## [Decision Letter · Decision Letter 0]

19 Dec 2022

PONE-D-22-29952Systems that support
hearing families with deaf children: a scoping
reviewPLOS ONE

Dear Dr. Terry,

Thank you for submitting your manuscript to PLOS ONE. After careful consideration, we
feel that it has merit but does not fully meet PLOS ONE’s publication criteria as it
currently stands. Therefore, we invite you to submit a revised version of the
manuscript that addresses the points raised during the review process.

Please submit your revised manuscript by Feb 02 2023 11:59PM. If you will need more
time than this to complete your revisions, please reply to this message or contact
the journal office at plosone@plos.org. When
you're ready to submit your revision, log on to https://www.editorialmanager.com/pone/ and select the 'Submissions
Needing Revision' folder to locate your manuscript file.

Please include the following items when submitting your revised
manuscript:A rebuttal letter that responds to each point raised by the academic
editor and reviewer(s). You should upload this letter as a separate file
labeled 'Response to Reviewers'.A marked-up copy of your manuscript that highlights changes made to the
original version. You should upload this as a separate file labeled
'Revised Manuscript with Track Changes'.An unmarked version of your revised paper without tracked changes. You
should upload this as a separate file labeled 'Manuscript'.

If you would like to make changes to your financial disclosure, please include your
updated statement in your cover letter. Guidelines for resubmitting your figure
files are available below the reviewer comments at the end of this letter.

We look forward to receiving your revised manuscript.

Kind regards,

Gursimran Dhamrait, Ph.D

Academic Editor

PLOS ONE

Journal Requirements:

"This scoping review and the linked descriptive qualitative study are part of the
SUPERSTAR project funded by Research Capacity Building Collaboration (RCBC) Wales,
which are part of a Postdoctoral Fellowship. The funder provides access to a
supervisor, and a Community of Scholars to support and promote high research quality
and outputs.  "

"J Terry  RCBC Wales Postdoctoral Fellowship

https://www.rcbcwales.org.uk/   

5. Please upload a copy of Figures 1 an 2, to which you refer in your text on page
9-10. If the figure is no longer to be included as part of the submission please
remove all reference to it within the text.

Reviewers' comments:

Reviewer's Responses to Questions

**Comments to the Author**

1. Is the manuscript technically sound, and do the data support the conclusions?

Reviewer #1: Yes

Reviewer #2: Yes

Reviewer #3: Yes

2. Has the statistical analysis been performed
appropriately and rigorously? 

Reviewer #1: Yes

Reviewer #2: N/A

Reviewer #3: N/A

3. Have the authors made all data underlying the
findings in their manuscript fully available?

Reviewer #1: No

Reviewer #2: Yes

Reviewer #3: Yes

4. Is the manuscript presented in an intelligible
fashion and written in standard English?

Reviewer #1: Yes

Reviewer #2: Yes

Reviewer #3: Yes

5. Review Comments to the Author

Reviewer #1: Thank you for the opportunity to review Systems that support hearing
families with deaf children: A scoping review. To help pinpoint the specific
sentences that require my commentary in the absence of line numbers, I will use page
numbers and the closest citation like this P:[X]

2:[4] - Support is not really defined here. I know it's not quite possible to define
support as this is an individual issue, but the paper claims that support is
lacking. What support would be needed?

3:[7,8] - Many people will find the term "developing countries" offensive. Perhaps
Global South? Formerly colonized countries?

3:[9] - List out these effects - perhaps cite Wyatte Hall's work on language
deprivation syndrome.

3:[13] - This sentence a bit both-sides the debate. As you are not the New York
Times, I don't think it's necessary to try to both-sides a clearly destructive
argument from the Oralists. All deaf children should have some exposure to signed
languages. Anything else is an enforcement of white supremacy values where only
spoken languages are valued and everything else can be discarded.

4:[17] - Let's be realistic. There is almost never signed alone. It is usually a
mixture, or spoken languages only.

5:[20] - Note that the width of the language development window is still under
debate. 5 years isn't a hard stop. Language acquisition continues up until teenage
years. But yeah. Early is good.

5:[21] - This next paragraph is very good. I think you can add more citations to
bolster your argument. Hall Hall and Caselli is good. I believe Lillo-Martin may
have a recent paper.

[22] - I don't think Geers et al is the right citation for this.

6:[28] - I've seen several quasi definitions of support but nothing that makes me
scream, yes, this is what support is. I would like to do this.

7:[30] - Very nice and clear. I like the use of OSF

8 - It's not immediately clear to me why educational databases were not searched as
this would cover Journal of Deaf Studies and Deaf Education, Deaf Education &
International, and the Annals of the Deaf

9 - I don't like the term hearing impaired but I wonder if using that term would net
more articles.

10 - Does Rayyan software require a citation?

11 - I think there's more funding to be honest. These are also Global North countries
with robust economies. I am not sure this is as much an interest issue as it is an
ability to receive funding.

11:[319 - I am not sure what reference 319 is, but I assume that will be fixed.
However, I don't think that signed/spoken/both is the correct distribution of
choices. It's usually both/spoken.

11: [19] - I am not sure that I agree that parents don't have time to engage with
deaf communities (love the use of plural here). I think the answer is that there's
usually not infrastructure to pipe parents to the appropriate deaf communities.

12:[18] - I want your point to be stronger here. Language does not interfere with
language. Language does not hurt any kind of learning. Even asking the question is
ableist.

12:[35] - I'm surprised no one mentioned ableist attitudes towards signed
languages.

13:[40] - Laura Mauldin's Made to Hear discusses some of this. Also mentions that
parents tend to think that their kids will be the successful ones that don't need
signed languages.

15:[45] - I think this kind of hints that communication will always be a joint
venture. I want you to say this stronger.

16:[50] - These families really should have an option to learn the signed languages
of their cultures (e.g. LSM)

17:[52] - I think the issue of socio-emotional development being poor isn't well
analyzed by hearing professionals. People react badly to not being able to
communicate and things like Dinner Table Syndrome. Like, it's not the deafness. It's
everything around it.

21:[69] - I do not understand this sentence "Aware that by giving parents ‘the means
to communicate with their deaf child by using the child’s primary language’ parents
are provided with greater opportunity for understanding their child’s world. "

21:[69] - This paragraph is rough. I don't understand what the point is. It doesn't
matter if the child overtakes the parent in proficiency. It only matters if the
parents are the only model, like, forever.

24:[77] - Without any context, reporting the data by race/ethnicity does not add
positive information. Why do Black families struggle with coping strategies? How
does the inherent racism of the US system restrict use of coping strategies? This
information isn't here. I would remove everything after "Additionally,"

25:[84] - Marschark is famous for never considering the context that the data is
gathered in. Why are these kids feeling lonely? What does it mean that kids
implanted earlier feel less lonely? Does anyone notice that the number of pirates
decreased as the weather got warmer (https://pastafarians.org.au/pastafarianism/pirates-and-global-warming/)?

25:[85] - The weirdness of conflated variables appears in Holt et al's work. I see
Kronenberger and Pisoni also contributed to this. This data result is silly. How
would self reported control lead to small vocabularies? How would organization mean
fewer inhibition problems? If a butterfly farts in the Atlantic, is there guaranteed
to be a hurricane in the Pacific?

26:[86] - Sad that most of the deaf clubs in the states have closed.

27:[91] - I would not cite Johnston here. I think Kuster's work is more applicable.
As is de Muelder.

30:[101] - Make sure you're citing Leala Holcomb's work here.

31:[40] - Honestly it's fair to say the hearing parents are ableist and audist
yeah.

31:[61] - In my US State, all the parent supporters are oralist advocates. We have no
signing parent mentor.

Reviewer #2: Suggestion:

p. 5, sentence beginning ‘Sign Language often comes naturally to deaf children…’ —
Rather than beginning the sentence with ‘Sign Language’ with both words capitalised,
I recommend using lower case in ‘language’ because there is no language named ‘Sign
Language’ but with both letters capitalized, it can create the impression that it is
the name of a language. This reinforces the common misconception that there is only
one sign language worldwide. By saying “sign language” with lowercase letters (or
‘signed languages’), it reads more parallel to ‘spoken language’ where no one would
mistake that for a single language but rather a category of languages in a
particular modality. This is done later in the MS (and even later in this paragraph)
and it reads much more clearly without this possibility of misunderstanding.

p. 11, first sentence under subheading — perhaps a typo in the citation. It is listed
as ‘[319’ with no closing bracket

Throughout: It seems you mostly chose to capitalise ‘deaf’, which is fine, however
this is a bit inconsistent, e.g., with some lowercase tokens on page 16. I’d suggest
explaining whatever decision you make, citing the relevant literature and then being
consistent throughout. If you’re not aware of this paper, I recommend it for a newer
take on ‘Deaf’ versus ‘deaf’

https://www.scirp.org/journal/paperinformation.aspx?paperid=97416

Positionality statement:

It would be helpful, if permitted by the journal, to include a positionality
statement for the two authors so readers know what lived experiences/expertise they
bring to this work. That would, to my mind, tie in with some of the narrative about
the findings reported in this paper.

Just in case it is helpful/applicable, please see Lieberman et al (2022)

https://sites.bu.edu/lavalab/files/2022/06/Lieberman-Mitchiner-Pontecorvo-2022.pdf

See also Oyserman and de Geus (2021)

https://www.degruyter.com/document/doi/10.21832/9781800410756-011/html

Reviewer #3: This paper is well-balanced and captures the nuance quite well of what
parents experience when they learn of having a deaf child and how the medical model
influences choices. I was impressed with the acknowledgement of how marginalized
signed languages have been in deaf child development. I applaud the authors for
bringing this balanced view and highlighting the plurality of options in an
international scope. I have no major feedback on this paper, there are minor grammar
and writing aspects that can be improved with editor review.

6. PLOS authors have the option to publish the peer
review history of their article (what does this mean?). If published, this will
include your full peer review and any attached files.

If you choose “no”, your identity will remain anonymous but your review may still be
made public.

**Do you want your identity to be public for this peer review?** For
information about this choice, including consent withdrawal, please see our
Privacy Policy.

Reviewer #1: **Yes: **Jon Henner

Reviewer #2: No

Reviewer #3: No

---

## [Author Response · Author response to Decision Letter 0]

5 Jan 2023

Response to reviewers – PONE – 22 – 29952 

Systems that support hearing families

We are grateful to all reviewers for their helpful comments. Please see our responses
below. On the tracked changes version of the manuscript, all new references in the
References list are highlighted in yellow. The numbering both in the manuscript and
in the reference list is now accurate for this revised manuscript.

Reviewer #1: Thank you for the opportunity to review Systems that support hearing
families with deaf children: A scoping review. To help pinpoint the specific
sentences that require my commentary in the absence of line numbers, I will use page
numbers and the closest citation like this P:[X]

Response: Thank you, your comments across the paper are extremely helpful.

2:[4] - Support is not really defined here. I know it's not quite possible to define
support as this is an individual issue, but the paper claims that support is
lacking. What support would be needed?

Response: Thank you. A definition of support is now included in the first paragraph
of the introduction.

3:[7,8] - Many people will find the term "developing countries" offensive. Perhaps
Global South? Formerly colonized countries?

Response Thank you, now amended to ‘the Global South’.

3:[9] - List out these effects - perhaps cite Wyatte Hall's work on language
deprivation syndrome.

Response: Thank you, effects and Wyatte Hall’s work now included.

3:[13] - This sentence a bit both-sides the debate. As you are not the New York
Times, I don't think it's necessary to try to both-sides a clearly destructive
argument from the Oralists. All deaf children should have some exposure to signed
languages. Anything else is an enforcement of white supremacy values where only
spoken languages are valued and everything else can be discarded.

Response: Helpful point, sentence with [13] now removed.

4:[17] - Let's be realistic. There is almost never signed alone. It is usually a
mixture, or spoken languages only.

Response: Thank you, now rephrased.

5:[20] - Note that the width of the language development window is still under
debate. 5 years isn't a hard stop. Language acquisition continues up until teenage
years. But yeah. Early is good.

Response: Thank you and a fair point, now softened with ‘around the age of five
years..’

5:[21] - This next paragraph is very good. I think you can add more citations to
bolster your argument. Hall Hall and Caselli is good. I believe Lillo-Martin may
have a recent paper.

Response: Very helpful, thank you, both suggested citations now added.

[22] - I don't think Geers et al is the right citation for this.

Response: Fair point, now removed.

6:[28] - I've seen several quasi definitions of support but nothing that makes me
scream, yes, this is what support is. I would like to do this.

Response: Thank you, good point. We have provided a definition of support systems at
this point.

7:[30] - Very nice and clear. I like the use of OSF

Response: Thanks.

8 - It's not immediately clear to me why educational databases were not searched as
this would cover Journal of Deaf Studies and Deaf Education, Deaf Education &
International, and the Annals of the Deaf

Response: Thanks. Whilst specific educational databases were not included as one of
the four named databases, many of the included 65 papers are indeed from educational
journals including Journal of Deaf Studies and Deaf Education, Deaf Education &
International, and the Annals of the Deaf. The Proquest Central database does
include educational databases. Also, due to the steering groups’ suggested search
terms, the information specialist’s help and intense internet and hand searching for
evidence the authors are confident that relevant evidence from educational journals
is certainly included in this scoping review. 

9 - I don't like the term hearing impaired but I wonder if using that term would net
more articles. 

Response: Agree, the term ‘hearing impaired’ was indeed used as a search term, please
see Table 1, PICO framework.

10 - Does Rayyan software require a citation?

Response: Thanks, good point. Rayyan reference (Ouzzani et al now included).

11 - I think there's more funding to be honest. These are also Global North countries
with robust economies. I am not sure this is as much an interest issue as it is an
ability to receive funding.

Response: Thanks, sentence now revised.

11:[319 - I am not sure what reference 319 is, but I assume that will be fixed.
However, I don't think that signed/spoken/both is the correct distribution of
choices. It's usually both/spoken.

Response: Thanks, typo now corrected. The reference had been Yu (2021), but now
changed to Hall, Hall and Caselli. Sentence now revised.

11: [19] - I am not sure that I agree that parents don't have time to engage with
deaf communities (love the use of plural here). I think the answer is that there's
usually not infrastructure to pipe parents to the appropriate deaf communities.

Response: Thanks, sentence now amended to reflect lack of infrastructure.

12:[18] - I want your point to be stronger here. Language does not interfere with
language. Language does not hurt any kind of learning. Even asking the question is
ableist.

Response: Thank you, sentence revised.

12:[35] - I'm surprised no one mentioned ableist attitudes towards signed
languages.

Response: Thank you. Agree, ableist attitudes are not specifically stated as such in
the literature, although that does appear to be what parent experience is
describing.

13:[40] - Laura Mauldin's Made to Hear discusses some of this. Also mentions that
parents tend to think that their kids will be the successful ones that don't need
signed languages.

Response: Thank you, reference to Maudlin’s work now included. 

15:[45] - I think this kind of hints that communication will always be a joint
venture. I want you to say this stronger.

Response: Thank you, good point, sentence now revised.

16:[50] - These families really should have an option to learn the signed languages
of their cultures (e.g. LSM)

Response: Thank you, a helpful addition and now included.

17:[52] - I think the issue of socio-emotional development being poor isn't well
analyzed by hearing professionals. People react badly to not being able to
communicate and things like Dinner Table Syndrome. Like, it's not the deafness. It's
everything around it.

Response: Thank you, additional sentence now included.

21:[69] - I do not understand this sentence "Aware that by giving parents ‘the means
to communicate with their deaf child by using the child’s primary language’ parents
are provided with greater opportunity for understanding their child’s world. "

Response: Thank you, sentence now revised.

21:[69] - This paragraph is rough. I don't understand what the point is. It doesn't
matter if the child overtakes the parent in proficiency. It only matters if the
parents are the only model, like, forever.

Response: Thank you, paragraph now revised.

24:[77] - Without any context, reporting the data by race/ethnicity does not add
positive information. Why do Black families struggle with coping strategies? How
does the inherent racism of the US system restrict use of coping strategies? This
information isn't here. I would remove everything after "Additionally,"

Response: Thank you, now revised and sentence removed.

25:[84] - Marschark is famous for never considering the context that the data is
gathered in. Why are these kids feeling lonely? What does it mean that kids
implanted earlier feel less lonely? Does anyone notice that the number of pirates
decreased as the weather got warmer (https://pastafarians.org.au/pastafarianism/pirates-and-global-warming/)?

Response: Thanks, additional sentence added.

25:[85] - The weirdness of conflated variables appears in Holt et al's work. I see
Kronenberger and Pisoni also contributed to this. This data result is silly. How
would self reported control lead to small vocabularies? How would organization mean
fewer inhibition problems? If a butterfly farts in the Atlantic, is there guaranteed
to be a hurricane in the Pacific?

Response: Thank you for drawing attention to this. We have returned to the paper and
revised the phrasing.

26:[86] - Sad that most of the deaf clubs in the states have closed.

Response: Thank you, sentence added and citations added.

27:[91] - I would not cite Johnston here. I think Kuster's work is more applicable.
As is de Muelder.

Response: Thank you. We have retained reference to the Johnston paper, as we are
referring to the retrieved paper from the included 65. We have supported this with a
citation from Kusters, DeMeulder and O’Brien to increase clarity.

30:[101] - Make sure you're citing Leala Holcomb's work here.

Response: Thank you, we have now added a citation on Holcomb’s work.

31:[40] - Honestly it's fair to say the hearing parents are ableist and audist
yeah.

Response: Thank you, point revised and made more clearly.

31:[61] - In my US State, all the parent supporters are oralist advocates. We have no
signing parent mentor.

Response: Thank you, agree this may vary by region hence no changes made, as point is
more about Deaf mentors, few Deaf professionals and supporting parents being a
welcome intervention.

Reviewer #2: 

Suggestion:

p. 5, sentence beginning ‘Sign Language often comes naturally to deaf children…’ —
Rather than beginning the sentence with ‘Sign Language’ with both words capitalised,
I recommend using lower case in ‘language’ because there is no language named ‘Sign
Language’ but with both letters capitalized, it can create the impression that it is
the name of a language. This reinforces the common misconception that there is only
one sign language worldwide. By saying “sign language” with lowercase letters (or
‘signed languages’), it reads more parallel to ‘spoken language’ where no one would
mistake that for a single language but rather a category of languages in a
particular modality. This is done later in the MS (and even later in this paragraph)
and it reads much more clearly without this possibility of misunderstanding.

Response: Thank you, very helpful, and now checked across the manuscript and only
appears as ‘sign language or ‘signed language’, unless at the start of a sentence or
in reference to a specific such as ASL or BSL.

p. 11, first sentence under subheading — perhaps a typo in the citation. It is listed
as ‘[319’ with no closing bracket

Response: Thank you, it was a typo, now amended.

Throughout: It seems you mostly chose to capitalise ‘deaf’, which is fine, however
this is a bit inconsistent, e.g., with some lowercase tokens on page 16. I’d suggest
explaining whatever decision you make, citing the relevant literature and then being
consistent throughout. If you’re not aware of this paper, I recommend it for a newer
take on ‘Deaf’ versus ‘deaf’

https://www.scirp.org/journal/paperinformation.aspx?paperid=97416

Response: Thank you, we were not previously aware of this paper, which is extremely
helpful and is now cited in our explanation. We have now included an Authors’ note
at the start of the paper to provide reader clarity. Throughout the paper all
reference to Deaf adults, communities, professionals, clubs, identity, culture,
mentors and family members, then a capital D is used. Whilst when referring to deaf
children, and deafness a lower-case d for deaf is used. 

Positionality statement:

It would be helpful, if permitted by the journal, to include a positionality
statement for the two authors so readers know what lived experiences/expertise they
bring to this work. That would, to my mind, tie in with some of the narrative about
the findings reported in this paper.

Response: Thank you, a useful point to consider. Positionality statements can be
useful for readers. We thought it helpful to add more description about the
project’s steering group in the Methods’ section. 

Just in case it is helpful/applicable, please see Lieberman et al (2022)

https://sites.bu.edu/lavalab/files/2022/06/Lieberman-Mitchiner-Pontecorvo-2022.pdf

See also Oyserman and de Geus (2021)

https://www.degruyter.com/document/doi/10.21832/9781800410756-011/html

Response: Very helpful to have these references. We have included them towards the
end of the paper, thank you for bringing them to our attention.

Reviewer #3: 

This paper is well-balanced and captures the nuance quite well of what parents
experience when they learn of having a deaf child and how the medical model
influences choices. I was impressed with the acknowledgement of how marginalized
signed languages have been in deaf child development. I applaud the authors for
bringing this balanced view and highlighting the plurality of options in an
international scope. I have no major feedback on this paper, there are minor grammar
and writing aspects that can be improved with editor review.

Response -Thank you for your comments.

to reviewers PONE-22-29952.docx
---

## [Decision Letter · Decision Letter 1]

26 Apr 2023

PONE-D-22-29952R1Systems that support
hearing families with deaf children: a scoping
reviewPLOS ONE

Dear Dr. Terry,

Thank you for submitting your manuscript to PLOS ONE. After careful consideration, we
feel that it has merit but does not fully meet PLOS ONE’s publication criteria as it
currently stands. Therefore, we invite you to submit a revised version of the
manuscript that addresses the points raised during the review process.

Please submit your revised manuscript by Jun 10 2023 11:59PM. If you will need more
time than this to complete your revisions, please reply to this message or contact
the journal office at plosone@plos.org. When
you're ready to submit your revision, log on to https://www.editorialmanager.com/pone/ and select the 'Submissions
Needing Revision' folder to locate your manuscript file.

Please include the following items when submitting your revised
manuscript:A rebuttal letter that responds to each point raised by the academic
editor and reviewer(s). You should upload this letter as a separate file
labeled 'Response to Reviewers'.A marked-up copy of your manuscript that highlights changes made to the
original version. You should upload this as a separate file labeled
'Revised Manuscript with Track Changes'.An unmarked version of your revised paper without tracked changes. You
should upload this as a separate file labeled 'Manuscript'.If you would like to make changes to your financial disclosure,
please include your updated statement in your cover letter. Guidelines for
resubmitting your figure files are available below the reviewer comments at the end
of this letter.

We look forward to receiving your revised manuscript.

Kind regards,

Gursimran Dhamrait, Ph.D

Academic Editor

PLOS ONE

Journal Requirements:

Additional Editor Comments:

1. Please re-word to use low-middle income countries and high-income countries as
appropriate throughout the manuscript. 

Reviewers' comments:

Reviewer's Responses to Questions

**Comments to the Author**

1. If the authors have adequately addressed your comments raised in a previous round
of review and you feel that this manuscript is now acceptable for publication, you
may indicate that here to bypass the “Comments to the Author” section, enter your
conflict of interest statement in the “Confidential to Editor” section, and submit
your "Accept" recommendation.

Reviewer #2: All comments have been addressed

2. Is the manuscript technically sound, and do the data
support the conclusions?

Reviewer #2: Yes

3. Has the statistical analysis been performed
appropriately and rigorously? 

Reviewer #2: Yes

4. Have the authors made all data underlying the
findings in their manuscript fully available?

Reviewer #2: Yes

5. Is the manuscript presented in an intelligible
fashion and written in standard English?

Reviewer #2: Yes

6. Review Comments to the Author

Reviewer #2: Overall, I’m very happy with the revisions to the manuscript. Thank you
for your hard work in addressing reviewer comments. I have just one more remark:

1. P. 53-54 (paragraph straddling these pages). Here you state that an alternative
may have been for these families to learn LSM, however, I’m looking at the original
article, and I’m only seeing that the families in this study were from
“Spanish-speaking” backgrounds. It does not specify their country (or territory,
e.g., Puerto Rico) of origin. I do not feel the suggestion to specifically name LSM
here is appropriate for two reasons. First, if the family is from, say, El Salvador,
why would they be compelled to learn LSM? Second, if the family lives in the US and
there is no one in their community who knows LSM, how is it functional for them to
learn a language they can’t use with anyone? I feel strongly the manuscript would be
better off with the sentence in question omitted.

I’m not sure if this would be helpful, but if you feel compelled to comment on some
of the unique struggles of families from a Spanish-speaking background, you may
consider this article:

Steinberg, A., Bain, L., Li, Y., Delgado, G., & Ruperto, V. (2003). Decisions
Hispanic Families Make After the Identification of Deafness. Journal of Deaf Studies
and Deaf Education, 8(3), 291–314. https://doi.org/10.1093/deafed/eng016

7. PLOS authors have the option to publish the peer
review history of their article (what does this mean?). If published, this will
include your full peer review and any attached files.

If you choose “no”, your identity will remain anonymous but your review may still be
made public.

**Do you want your identity to be public for this peer review?** For
information about this choice, including consent withdrawal, please see our
Privacy Policy.

Reviewer #2: No

---

## [Author Response · Author response to Decision Letter 1]

1 May 2023

PONE-D-22-29952R1

We are grateful to all reviewers for their helpful comments. Please see our responses
below, and on the tracked changes version of the manuscript:

Editor comments: 

1. Please re-word to use low-middle income countries and high-income countries as
appropriate throughout the manuscript. 

Response: Thank you for this helpful comment. Now changed throughout. 

Reviewer #2: Overall, I’m very happy with the revisions to the manuscript. Thank you
for your hard work in addressing reviewer comments. I have just one more remark:

1. P. 53-54 (paragraph straddling these pages). Here you state that an alternative
may have been for these families to learn LSM, however, I’m looking at the original
article, and I’m only seeing that the families in this study were from
“Spanish-speaking” backgrounds. It does not specify their country (or territory,
e.g., Puerto Rico) of origin. I do not feel the suggestion to specifically name LSM
here is appropriate for two reasons. First, if the family is from, say, El Salvador,
why would they be compelled to learn LSM? Second, if the family lives in the US and
there is no one in their community who knows LSM, how is it functional for them to
learn a language they can’t use with anyone? I feel strongly the manuscript would be
better off with the sentence in question omitted.

I’m not sure if this would be helpful, but if you feel compelled to comment on some
of the unique struggles of families from a Spanish-speaking background, you may
consider this article:

Steinberg, A., Bain, L., Li, Y., Delgado, G., & Ruperto, V. (2003). Decisions
Hispanic Families Make After the Identification of Deafness. Journal of Deaf Studies
and Deaf Education, 8(3), 291–314. https://doi.org/10.1093/deafed/eng016

Response: Thank you for this helpful comment. On consideration we have removed this
sentence.

to reviewers PONE-D-22-29952R1.docx
---

## [Decision Letter · Decision Letter 2]

19 Jun 2023

PONE-D-22-29952R2Systems that support
hearing families with deaf children: a scoping
reviewPLOS ONE

Dear Dr. Terry,

Thank you for submitting your manuscript to PLOS ONE. After careful consideration, we
feel that it has merit but does not fully meet PLOS ONE’s publication criteria as it
currently stands. Therefore, we invite you to submit a revised version of the
manuscript that addresses the points raised during the review process.

Please check your prisma flow diagram in figure 1and explain exclusion criteria
inside the box. There is a need of improvement in PRISMA FLOW fig 1. Please see page
13. Justify exclusion screening (n=821). The diagram have made poorly. The line are
irregular. Kindly redraw and elaborate in detail all criteria. Please do include it
in text if any new addition in text.

Table 2, Strength and weakness must possess under a separate column. Rewrite in and
elaborate the content.

i could not find the search strategy as additional file. Kindly check your
manuscript. It is suggested to provide inside the manuscript.

Please write your manuscript according to plosone criteria and provide following
clear methodology in method section such as Data screening, data entry and
collection method.

read following content in Cochrane database

https://www.cochranelibrary.com/about/about-cochrane-reviews

Please submit your revised manuscript by Aug 03 2023 11:59PM. If you will need more
time than this to complete your revisions, please reply to this message or contact
the journal office at plosone@plos.org. When
you're ready to submit your revision, log on to https://www.editorialmanager.com/pone/ and select the 'Submissions
Needing Revision' folder to locate your manuscript file.

Please include the following items when submitting your revised manuscript:A rebuttal letter that responds to each point raised by the academic
editor and reviewer(s). You should upload this letter as a separate file
labeled 'Response to Reviewers'.A marked-up copy of your manuscript that highlights changes made to the
original version. You should upload this as a separate file labeled
'Revised Manuscript with Track Changes'.An unmarked version of your revised paper without tracked changes. You
should upload this as a separate file labeled 'Manuscript'.If you would like to make changes to your financial disclosure,
please include your updated statement in your cover letter. Guidelines for
resubmitting your figure files are available below the reviewer comments at the end
of this letter.

We look forward to receiving your revised manuscript.

Kind regards,

Muhammad Shahzad Aslam, Ph.D.,M.Phil., Pharm-D

Academic Editor

PLOS ONE

Journal Requirements:

Additional Editor Comments:

Please check your prisma flow diagram in figure 1and explain exclusion criteria
inside the box. There is a need of improvement in PRISMA FLOW fig 1. Please see page
13. Justify exclusion screening (n=821). The diagram have made poorly. The line are
irregular. Kindly redraw and elaborate in detail all criteria. Please do include it
in text if any new addition in text.

Table 2, Strength and weakness must possess under a separate column. Rewrite in and
elaborate the content.

i could not find the search strategy as additional file. Kindly check your
manuscript. It is suggested to provide inside the manuscript.

Please write your manuscript according to plosone criteria and provide following
clear methodology in method section such as Data screening, data entry and
collection method.

read following content

https://www.cochranelibrary.com/about/about-cochrane-reviews

Reviewers' comments:

Reviewer's Responses to Questions

**Comments to the Author**

1. If the authors have adequately addressed your comments raised in a previous round
of review and you feel that this manuscript is now acceptable for publication, you
may indicate that here to bypass the “Comments to the Author” section, enter your
conflict of interest statement in the “Confidential to Editor” section, and submit
your "Accept" recommendation.

Reviewer #2: All comments have been addressed

2. Is the manuscript technically sound, and do the data
support the conclusions?

Reviewer #2: Yes

3. Has the statistical analysis been performed
appropriately and rigorously? 

Reviewer #2: Yes

4. Have the authors made all data underlying the
findings in their manuscript fully available?

Reviewer #2: Yes

5. Is the manuscript presented in an intelligible
fashion and written in standard English?

Reviewer #2: Yes

6. Review Comments to the Author

Reviewer #2: I have no further comments at this time. I am satisfied with the way the
manuscript has been revised.

7. PLOS authors have the option to publish the peer
review history of their article (what does this mean?). If published, this will
include your full peer review and any attached files.

If you choose “no”, your identity will remain anonymous but your review may still be
made public.

**Do you want your identity to be public for this peer review?** For
information about this choice, including consent withdrawal, please see our
Privacy Policy.

Reviewer #2: No

---

## [Author Response · Author response to Decision Letter 2]

28 Jun 2023

Response to Reviewers- - PONE-D-22-29952R2

Thank you for your helpful comments and guidance. As we have revised the manuscript,
we have looked extensively at other scoping reviews published in Plos One, which has
been helpful.

Reviewer comment: Please check your prisma flow diagram in figure 1and explain
exclusion criteria inside the box. There is a need of improvement in PRISMA FLOW fig
1. Please see page 13. Justify exclusion screening (n=821). The diagram have made
poorly. The line are irregular. Kindly redraw and elaborate in detail all criteria.
Please do include it in text if any new addition in text.

Response: Thank you for your suggestions. The PRISMA diagram has been re-done. No
further text added to manuscript.

Reviewer comment: Table 2, Strength and weakness must possess under a separate
column. Rewrite in and elaborate the content.

Response: Separate columns have now been added for strengths and another for
weaknesses, with additional content now added. A few entries do just state ‘small
sample’, and we have been back to these articles again, but there is little else to
say with regards to these publications due to limited detail.

Reviewer comment: I could not find the search strategy as additional file. Kindly
check your manuscript. It is suggested to provide inside the manuscript.

Response: Apologies. An example of a search strategy from one database (CINAHL) is
now included within the manuscript and has also been uploaded as an additional file. 

Reviewer comment: Please write your manuscript according to Plosone criteria and
provide following clear methodology in method section such as Data screening, data
entry and collection method.

Response: Thank you. Further detail about data screening, data entry and collection
method is now included on page 12. The sub-headings used in the Methods sections are
based on Arksey and O’Malley’s framework, as explained on page 8.

to Reviewers_PONE-D-22-29952R2 .docx
---

## [Editor Report · Decision Letter 3]

4 Jul 2023

Systems that support hearing families with deaf children: a scoping review

PONE-D-22-29952R3

Dear,

We’re pleased to inform you that your manuscript has been judged scientifically
suitable for publication and will be formally accepted for publication once it meets
all outstanding technical requirements.

Kind regards,

Muhammad Shahzad Aslam, Ph.D.,M.Phil., Pharm-D

Academic Editor

PLOS ONE
---

## [Editor Report · Acceptance letter]

11 Jul 2023

PONE-D-22-29952R3 

Systems that support hearing families with deaf children: a scoping review 

Dear Dr. Terry:

I'm pleased to inform you that your manuscript has been deemed suitable for
publication in PLOS ONE. Congratulations! Your manuscript is now with our production
department. 

Kind regards, 

on behalf of

Dr. Muhammad Shahzad Aslam 

Academic Editor

PLOS ONE